# Practices of Motivators in Adopting Agile Software Development at Large Scale Development Team from Management Perspective

**Rashid Ali Khan** [1][ID]**, Muhammad Faisal Abrar** [2][ID]**, Samad Baseer** [3]**, Muhammad Faran Majeed** [4][ID]**, Muhammad Usman** [5][ID]**, Shams Ur Rahman** [2] **and You-Ze Cho** [6,*][ID]

1   Department of Computer Science, Qurtuba University, Peshawar 25000, Pakistan; rashid.buniry@gmail.com
2   Department of Computer Science, University of Engineering and Technology, Mardan 23200, Pakistan; abrarfaisal49@gmail.com (M.F.A.); shams@uetmardan.edu.pk (S.U.R.)
3   Department of Computer System Engineering, University of Engineering and Technology, Peshawar 25000, Pakistan; drsamadbaseer@uetpeshawar.edu.pk
4   Department of Computer Science, Shaheed Benazir Bhutto University, Sheringal 18000, Pakistan; m.faran.majeed@ieee.org
5   Department of Computer Software Engineering, University of Engineering and Technology, Mardan 23200, Pakistan; usman@uetmardan.edu.pk
6   School of Electronics Engineering, Kyungpook National University, Daegu 41566, Korea
*   Correspondence: yzcho@ee.knu.ac.kr

**Abstract:** Agile software development methodologies have become the most popular software development methods in the last few years. These methodologies facilitate rapid development. The low cost and prioritized user satisfaction make these methodologies more attractive. These methodologies were also intended for small scale developmental teams. Therefore, challenges were encountered when these methodologies were used in large-scale development teams. This study was based on the identification of factors which were discovered in our previous study. Some of the factors included "leadership strong commitment and team autonomy", "cooperative organizational culture", and "team competency—agile development expertise". A total of 147 practices were identified in this study via a systematic literature review. These practices will help practitioners and project managers to adopt agile software methodologies and encourage them to the enhance them.

**Keywords:** practices; systematic literature review; agile transformation; scaling agile

## 1. Introduction and Background

"Agile methods" have been around for two decades, and standards and common practices related to agile software development exist. However, there is still no complete agreement on what exactly agile software development methods are. Agile software development methods are used to develop or extend a software packages when the requirements keep changing. The flexibility of the agile methods allows one to cope with requirement volatility and facilitates close teamwork among clients and programmers, making agile software development highly attractive. Agile methods enable the development of software to take place rapidly. They involve non-stop code assimilation and have the capacity to tackle altering business requirements [1]. These methods allow the clients to frequently request additions of new features. A well-known example of agile methods is extreme programming (XP). XP involves multiple short development cycles, rather than a long one; each cycle involves the activities of coding, testing, listening, and designing. The listening activity involves obtaining, updating, or revising requirements from the customer [2]. For small-scale software projects, agile methods may deliver the final products more quickly than other software development methods. Skepticism naturally arises, from the management perspective, regarding using agile methods for large-scale projects. One

might ask whether agile methods can be used to develop large-scale software products and what practices should be adopted for successful completion of large-scale projects using agile methods? Agile software development started in 2001 with what is called VersionOne, which is used for small-scale projects. With the passage of time, large-scale software applications begun to be developed by agile methods, as they adopted scaling software development techniques. Thus, agile methods are now used by large software development teams [3,4]. In a recent survey involving four thousand participants, 62% had more than 100 developers in their software houses. In addition to this, 43% of the participants stated that they use agile. However, for a large-scale software development project using agile methods to succeed, certain factors called success factors must be taken care of [5].

The concept of success factors (SFs) was presented by D. Ronald Daniel [6] and was further refined by John F. Rockart between 1979 [7] and 1981 [8], who stated that SFs are the factors to which management must give constant and careful attention. Once SFs are identified, key performance indicators can be developed for performance assessments [9]. According to the authors of [9,10], SFs are the components needed to build an environment where the given project can be managed in the best possible way. The main SF of a project is customer satisfaction. Other factors include "traditional organizational culture", "team collaboration", and "project management" [11]. Hence, SFs may be viewed as the ways in which certain businesses operate in the right way and then succeed. However, SFs also depend on the management's concerns; if they do not pay attention to the critical factors, then the results can be bad, and the organization may suffer [12]. Therefore, managers must focus on SFs. Over time, SFs may vary depending on the position of the employee in an organization or geographic location [12,13]. In the context of agile methods, SFs (or motivators) can be defined as "the factors leading towards the successful adoption of agile software development methods (ASDM) for large-scale projects involving large teams". The right practices are the solutions to the SFs (motivators). In this paper, we use the terms SFs and motivators interchangeably. This paper is about the practices that focus on success factors within agile software development methodologies, made for large-scale development, from the management perspective. The size of a project depends on the number of people involved, the number of teams involved, the total amount of code involved, the project budget, etc. The author of [12] stated that a large-scale project involves at least seven teams with forty people in total, whereas the authors of [14] argued that a large-scale project should have a budget of at least ten million GBP and involves at least fifty people. The author of [1] stated that only projects with over five million lines of code can be called large projects. Moreover, the authors of [9] said that the duration of a large-scale project should be at least two years. According to the author of [13], in a large-scale project there should be two to nine teams collaborating with one another, and anything above nine collaborating teams is a very large-scale project.

Hence, it is evident that disagreement exists among the studies undertaken to find large-scale agile development (LSAD) teams and projects. For example, while some studies, such as [15,16], categorized projects involving 50 people as large-scale, other studies, such as [14], considered the manpower of 50 people to be small and only considered a project as large-scale if the number of people working on it was more than 70. As for the nature of the workforce, there is no restriction that all employees working on the project should be programmers, but they will need to collaborate when required. Organization size can be found by using scrum masters and software architects. Business needs and management functions are also met by agile methodology. Every organization should move forward to the goal of being iterative and must be open to new functionality-based models. Following the software development life cycle (SDLC) should be avoided. Concentration is required on the short-term project-level planning [16]. Agile methodology is focused on planning, and its creators argue that it will be the future of development, but there are methodological problems and room for oversights concerning customer–company relationships. For the purpose of short-term planning, the mobility of investors needs to be involved in both

operation and analysis phases. Our aim was to fill the gap in research regarding projects and teams for LSAD from a management perspective, with a systematic literature review (SLR). To the best of our knowledge, no SLR exists that explored the relationship between ASDM and large-scale development. We found certain factors in our previous study [17] which have significant impacts on the adoption of ASDM for large-scale developmental team from management perspectives.

For this purpose, we drafted a research question (RQ) related to the literature review.

**RQ1** What are the solutions/practices in ASDM used to address motivators in large-scale situations from a management perspective?

### 2. Research Methodology

In order to identify the methodologies that can be used for the proper implementation of success factors or motivators, we performed a systematic literature review (SLR) [18]. The research methodology is presented in Figure 1.

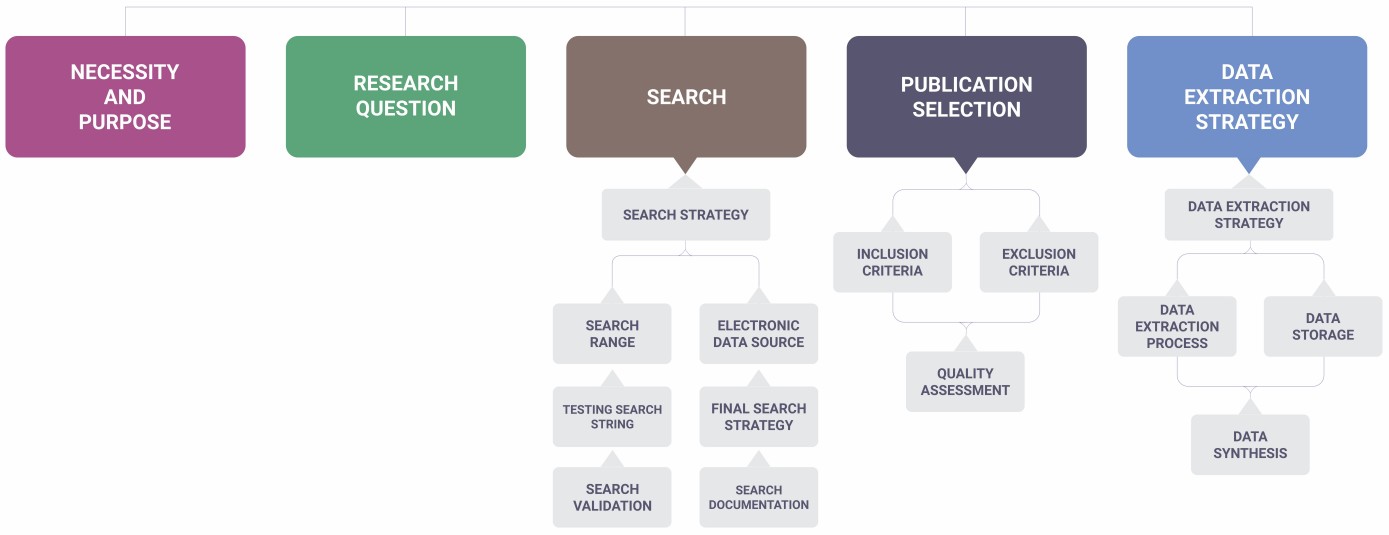

**Figure 1.** SLR process.

Using the SLR, we aimed to discover the practices used to achieve in terms of the success factors identified in our previous study [17], in large-scale ASDM. The search strategy in the SLR covered all the possible publications: conferences, precedings, international journal papers, reports case studies, theses, dissertations, etc. Our study should provide the right directions and approaches to project managers aiming to adopt ASDM. The novelty of our research lies in the fact that currently no other SLR exists for ASDM for large-scale projects.

We will help to improve the readiness of project managers, developers, and team members using ASDM. The ultimate goal of our project—scaling agile development—is underway. Our paper indicates the most suitable practices for each success factor. The following steps were derived from the guidelines for performing SLRs in software engineering [18].

1. Define the research objective.
2. Conduct several example searches; review the scopes.
3. Define the search string; identify inclusion and exclusion criteria.
4. Conduct an initial search.
5. Review the title, abstract, and keywords of the initially retrieved studies.
6. Revise inclusion and exclusion criteria; select potentially relevant studies.
7. Remove duplicate studies.
8. Review potentially relevant studies selected; discuss any issues.

9. Review the entire contents of initially selected studies (including the references section for identifying studies that were potentially missed); identify relevant ones.
10. Review relevant studies selected; discuss any issues.
11. Identify the final set of relevant studies.

Figure 2 starts from step 4, as we depict the publication selection process therein. The major steps are explained below.

### 2.1. Search Strategy

Manual searching in the digital libraries (IEEE, Science Direct, ACM, CiteSeer, and Springer) was conducted with the help of a search string.

### 2.1.1. Search String

We used search strings in different digital libraries, such as IEEE, ACM, and Springer-Link, as each of these libraries accepts a different size of string. Some of them accept large strings, and some of them only accept small strings. The first step was to trial a search string. This is explored in the subsequent section.

### 2.1.2. Trial Search String

We used the search strings given below to guide the baseline search in the Science Direct Digital Library. If the result of the search string was accurate, the result was used. Examples of search strings are ("Agile Methodology") and ("Large Development Team") and (Success Factors or Motivators).

### 2.1.3. Lengthy Search String

We created search strings by combining major terms and their synonyms using Boolean operators. If the database source (digital libraries) allowed a given string and provided results, it was a search string, otherwise we divided it into smaller sub-strings.

### 2.1.4. Smaller Search Sub-Strings

Due to the fact that some libraries do not allow large strings, we divided the search string into smaller sub-strings by using Boolean operations (AND/OR) and obtained the results. To remove the repetitions, we summarize the search outcomes. For the construction of search terms we used the following steps.

**Step 1:** Derivation of Major Terms
To guide key terms, perceive population, intervention, and outcome from research questions.
**Step 2:** Identification of Alternative Spellings and Synonyms
Found the alternative spellings and synonyms for these major terms.
**Step 3:** Verification of Keywords
Validated the key words in any related paper.
**Step 4:** Use of Boolean Operators for Conjunction
The Boolean operators (AND/OR) were used to combine search segments which were related to our research topic. The "OR" operator was used for combing alternative spellings and synonyms, and the "AND" operator was used for combing major terms.

The final large search string was as follows. ("Agile Method" OR "Agile Software Development" OR "Agile Method OR Agile Development") AND ("Large Development Team" OR "Large Development Team" OR "Large Development Team") AND ("Incentives" OR "motivators" OR "Factors" OR "Success The factor" OR "positively affects" OR "promoters" OR "supporters" OR "key factors") AND ("Practices" OR "Solutions").

As some libraries do not allow large strings, we divided the search string into smaller search strings as follows.

**String1:** ("Agile technology OR agile methods") AND ("large-scale development teams") OR ("motivators" OR "positive impact") AND ("Practices" OR "Solutions").
**String2:** ("Agile Method") AND ("Development Team") AND ("Success Factors" OR "Positive Impacts" OR "Promoters") AND ("Practices" OR "Solutions").
**String3:** ("Agile Software Development" OR "Agile System Development") AND ("Large Development Team") AND (factors OR supporters OR "key factors") AND ("Practices" OR "Solutions").

### 2.2. Publication Selection

Inclusion and Exclusion Criteria

The selection of papers is a hectic task. To select relevant papers and to exclude irrelevant papers, we adopted inclusion and exclusion criteria. The inclusion criteria were that the paper was in English, and that the keywords in the title of the paper and the keywords in our research matched. Our stress was on the matching of the keywords/major terms that appeared as a result of the search string. These results provided a list of publications. The first step was to read each paper's title and apply the inclusion/exclusion criteria. If the paper's title matched with major terms of our research question, then the abstract was read for confirmation of the relevance by applying the inclusion criteria. Contrary to this, if the search string results did not match even a little with the major terms in our research question, then we applied exclusion criteria. The inclusion and exclusion criteria can bee seen in Table 1.

**Table 1.** Inclusion/Exclusion Criteria.

| Inclusion Criteria | Exclusion Criteria |
| --- | --- |
| Papers that refers to practices for the Success Factors in the adoption of ASDM at large scale from management perspectives. | Papers that are not related to the Research Question. |
| Papers that are transcribed in English only and full text is available. | Papers that do not follow inclusion criteria. |

### 2.3. Selecting Primary Sources

The selection of primary sources consisted of two parts. In the initial phase, the selection was done by briefly reading the title and abstract of each paper, which was then followed by the final selection in which we read the entire scientific articles. We also conducted an inter-rater reliability test to remove the bias, but we found none.

Only 95 papers out of the selected 366 met the inclusion criteria. Furthermore, we removed 13 papers due to duplication in different libraries. In the end we had a final count of 82 papers. Figure 2 depicts the number of manuscripts selected at steps 8–11.

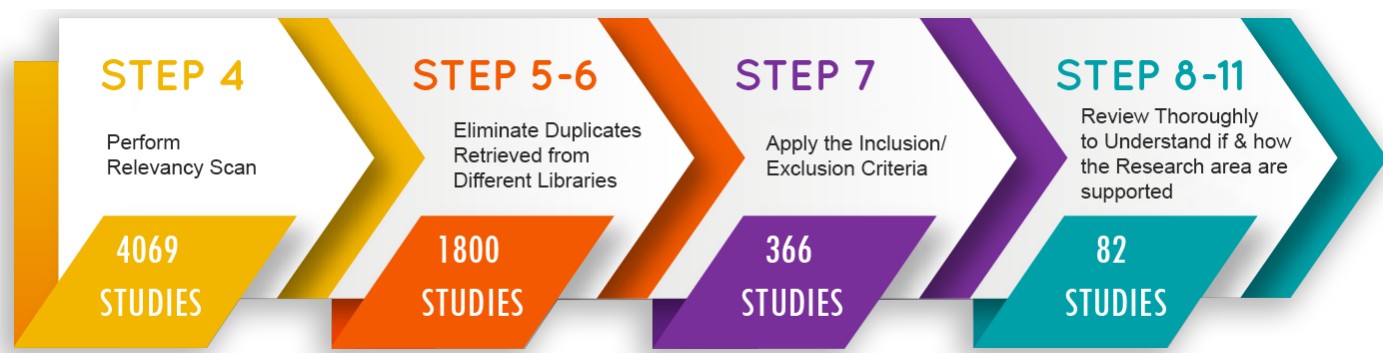

**Figure 2.** Publication selection process.

*2.4. Publication Quality Assessment*

After applying the quality assessment criteria to the results obtained in step 8 (see Figure 2), 82 papers qualified for the final list. The principle task of a quality assessment is to check and evaluate the nature of the papers. The quality checklist contained the following questions:

- Is it clear in what way the solution or practices for the success factors in the adoption of ASDM at a large scale were identified?
- Are there adequate data to support the results?
- Is the researcher reporting on favorable results more than adverse results?
- Is the objective of the research clearly defined?
- Are the outcomes of the research connected to the objective of the research?
- Is the ASDM context discussed clearly?

Each of the above answers was marked as yes, no, or N.A. Each paper was scored depending on the answers given to the questions. Each question represented 1 point. The minimal score needed to pass the quality assessment was 50%. The first question was mandatory. If this question was a no and the rest of the questions were answered correctly, then the publication still failed the quality assessment. All of the 82 selected papers achieved the minimal score, and thus the data were extracted accordingly.

*2.5. Data Extraction*

After extraction of data, no disagreement was found, even after the inter-rater reliability test. The following data were extracted from each scientific article published: date of review, title of the paper, authors, references, databases, practices, methodology used in paper (interview, case study, reports, or survey) target population, country, and location of research.

*2.6. Data Analysis*

We used frequency analysis for our data. All of the agile practices were counted. The list of these practices is shown in Appendix B. The relative significance was found by comparing it with the remaining agile practices. The classification of the final set of papers was performed with categories based on the practices of the success factors for the adoption of ASDM at large scales. All this information is shown in Appendix A from the final selected papers. The selected publications were classified on the basis of methodology: SLR, case study, or other methods. The information was extracted based on the authors' affiliations and the study strategies used for the research. Figure 3 depicts the high frequency of case studies. This analysis shows details of the selected publications and their methods.

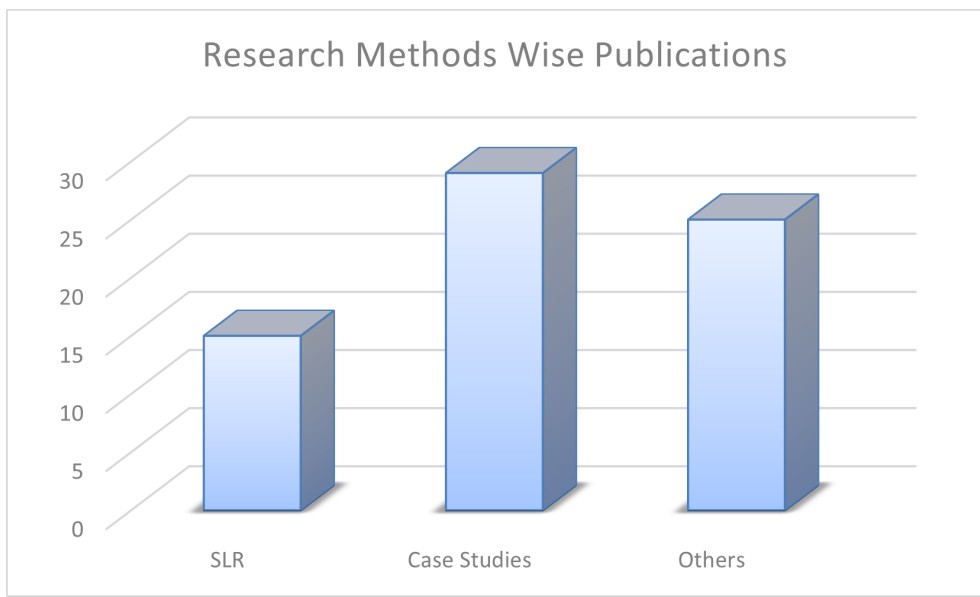

**Figure 3.** Publications by type of research.

We classified the selected papers based on location (continent) and method-wise to analyze the adoption of agile practices for large-scale projects in each continent. In this regard, the information was extracted based on countries of the authors or where the case study data were collected. Figure 4 illustrates the continent-wise frequency details of the selected publications.

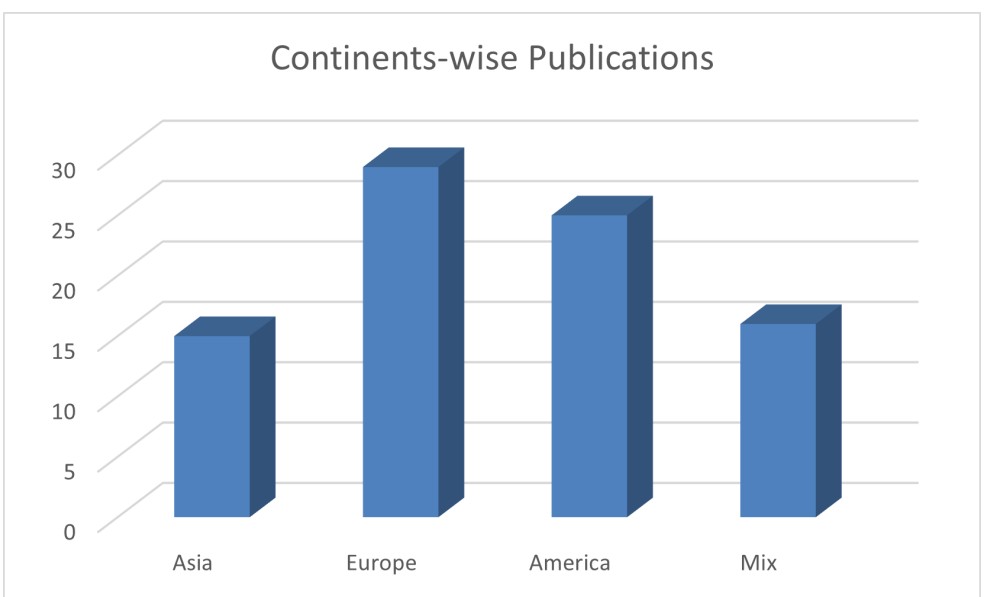

**Figure 4.** Publications by continent.

*2.7. Classification of Practices*

After identifying the practices for proper implementation of SFs in scaling ASDM SLR, we classified a few practices as critical practices. The classification of critical practices was based upon criteria such as the following: those practices were considered as critical which appeared in $\geq 15$ , 20, 35, or 30 papers. Removing the thresholds or making it a single standard value would drastically increase the number of unrelated practices. Therefore, we used different thresholds to find the important practices.

## 3. Results and Discussion

This section demonstrates the outcomes of the SLR. The research question established was to find out those solutions/practices from the literature using SLR which should provide significant help to software project managers in the adoption of ASDM at large scales. This section further explains this RQ in detail, alongside proper results shown in tables, as bullet points, and as graphs. In the following tables, we have used the term "motivators" for short to represent success factors. The subsequent sections reveal 21 motivators, as given in Table 2. Their respective practices are given in Tables 3–21.The paper IDs in the fifth column in Table 2 were actually taken from the list of finally selected papers, as mentioned in Appendix C.

**Table 2.** Success factors identified via SLR [17].

| S. No. | Success Factors | Frequency out of 58 | Percentage | Paper ID |
|---|---|---|---|---|
| 1 | Leadership strong commitment and team autonomy | 26 | 45% | P1, P2, P4, P5, P6, P7, P8, P9, P11, P12, P13, P14, P15, P17, P21, P22, P23, P24, P25, P26, P27, P28, P29, P51, P54, P57 |
| 2 | Cooperative organizational culture | 25 | 44% | P3, P5, P10, P16, P18, P19, P20, P21, P30, P31, P32, P33, P34, P35, P36, P37, P38, P39, P40, P41, P42, P43, P44, P45, P46 |
| 3 | Team competency—agile development expertise | 23 | 40% | P8, P9, P10, P16, P18, P19, P20, P21, P30, P31, P32, P33, P34, P35, P36, P37, P38, P39, P40, P47, P48, P49, P50 |
| 4 | Training and learning and briefing of top management on agile | 23 | 40% | P1, P2, P3, P6, P8, P14, P19, P21, P30, P31, P33, P34, P35, P36, P37, P51, P52, P53, P54, P55, P56, P57, P58 |
| 5 | Customer satisfaction | 21 | 37% | P7, P13, P15, P16, P19, P21, P30, P31, P33, P34, P35, P36, P37, P41, P42, P43, P44, P45, P46, P47, P48 |
| 6 | Strong collaboration with customer | 21 | 37% | P3, P13, P15, P16, P19, P21, P30, P31, P33, P34, P35, P36, P37, P41, P42, P43, P44, P45, P46 |
| 7 | Sustainable planning | 19 | 33% | P9, P11, P12, P16, P19, P21, P30, P31, P33, P34, P35, P36, P37, P41, P42, P43, P44, P45, P46 |
| 8 | Requirements management using agile-oriented requirement management process | 18 | 32% | P11, P12, P16, P19, P21, P30, P31, P33, P34, P35, P36, P37, P41, P42, P53, P54, P55, P56 |
| 9 | Use of automated software tools | 15 | 26% | P19, P21, P30, P31, P33, P34, P35, P36, P37, P41, P42, P43, P54, P55, P58 |
| 10 | Scheduled trainings for team members | 15 | 26% | P20, P21, P30, P31, P33, P34, P35, P36, P37, P41, P42, P43, P54, P55, P58 |
| 11 | Strong collaborations and communications | 14 | 25% | P21, P29, P21, P23, P24, P35, P36, P37, P41, P42, P43, P44, P45, P48 |
| 12 | Face-to-face meetings | 13 | 23% | P19, P21, P23, P24, P25, P26, P37, P41, P42, P43, P44, P25, P47 |
| 13 | Strong executive support | 12 | 21% | P21, P23, P24, P25, P26, P37, P41, P42, P43, P54, P55, P57 |

**Table 2.** *Cont.*

| S. No. | Success Factors | Frequency out of 58 | Percentage | Paper ID |
|---|---|---|---|---|
| 14 | Risk management | 12 | 21% | P11, P13, P24, P25, P26, P37, P41, P42, P43, P44, P45, P47 |
| 15 | Mechanism for change management | 8 | 14% | P25, P26, P37, P41, P42, P43, P44, P45 |
| 16 | Knowledge sharing management | 7 | 13% | P25, P26, P27, P32, P33, P44, P55 |
| 17 | Quality production using pair programming | 7 | 13% | P15, P16, P27, P32, P43, P44, P51 |
| 18 | Dedicated management | 6 | 11% | P16, P17, P22, P33, P44, P58 |
| 19 | Pilot project in case of no experience | 5 | 9% | P19, P22, P25, P31, P47 |
| 20 | Agile development environment | 3 | 6% | P11, P25, P27 |
| 21 | Team encouragement | 3 | 6% | P21, P25, P34 |

We have identified 147 practices in total, for achieving SFs or motivators which should lead project managers to successful large scale adoption of ASDM.

The software development businesses will also benefit from these practices by knowing how can they solve problems in ASDM at a large scale with conventional developmental methods.

### 3.1. Strong Executive Support

The organization running a project has a major impact on the completion of that project, and different researchers have shown that organizational factors are also important factors for success in large-scale agile projects [19–21]. These factors are further addressed, and include customer obligations, conclusion time, team delivery, company culture, planning and control, and business criticality [15,22,23].

We have found five practices critical (% ≥ 30) for implementing the SF "strong executive support". They are shown in Table 3.

- Manage the product and separate quality assurance groups.
- For a team to be effective and successful, all team members should be able to adapt to the ever changing requirements in the task environment.
- The ability to easily respond to change requirements of the customer.
- The team members must support each other during the whole work flow of the project.
- Trust building among the members of the agile team help the team function as one unit.

### 3.2. Cooperative Organizational Culture

For governing of the success of an agile project, a cooperative environment and test coverage quality are considered to be the most important categories [24–26]. A cooperative culture mainly focuses on humans working in pairs; the concepts of scopes and team scope, along with employee experience, are important. Risk aversion is also a helpful factor which helps us in strict planning and monitoring of the team. The success factors which have been identified in organizations from the literature review have been classified as upper management support, project planning monitoring, and change management [4,16].

**Table 3.** Practices for implementing strong executive support.

| S. No. | Practices for Implementing Strong Executive Support Identified through SLR | % of Practices Identified through SLR |
|:---:|---|:---:|
| 1. | Manage the product and separate quality assurance groups. | 36% |
| 2. | For any successful and effective team the members need to be adaptable to any change in the task environment. | 34% |
| 3. | The ability to easily respond to change requirements of the customer. | 30% |
| 4. | The team members must be support each other for executing Work flow of the project. | 32% |
| 5. | The members of the agile team should be mature in trust building to make it function as one unit. | 32% |

Table 4 states that a cooperative organizational culture can be best established by adopting the following seven critical practices (% ≥ 25):

- Focus on the importance of the culture and understand the main challenge to successful agile development on a large scale.
- Mutual understanding can be achieved by having active cross-cultural communication programs through the use of short visits and face-to-face communication.
- There are differences in norms and values among cultures.
- Be near the client for better communication.
- Better use of middlemen can reduce the communication barrier with clients.
- Time management techniques should be used to understand different time zones.
- A common set of development tools and policies should be used to facilitate common understanding.

For implementing the "cooperative organizational culture" SF, our SLR study founds 11 practices.

**Table 4.** Practices for implementing cooperative organizational culture.

| S. No. | Implementing the Practices of Cooperative Organizational Culture by SLR | % of Practices Identified through SLR |
|:---:|---|:---:|
| 1. | Briefly move the selected programmers to the Clients site. | 24% |
| 2. | Differences among other departments of the organizations should be reduced. | 23% |
| 3. | Focus on the importance of the Culture and understand what is main blockage in the successful migration to the Agile development on a large scale. | 28% |
| 4. | Mutual understanding can be achieved by having active cross cultural communication programs through the use of short visits and face-to-face communication. | 30% |
| 5. | Different skill training which can be a combination of formal communication languages, client specific requirements, or either domain specific along with logical thinking. | 24% |
| 6. | There will be difference in Norms and Values from other cultures. | 29% |
| 7. | Move to the clients site for better communication. | 32% |
| 8. | Better use of middlemen can be made to adjust the communication barrier among clients. | 25% |
| 9. | Time management techniques should be use to understand the different time zones. | 30% |

**Table 4.** *Cont.*

| S. No. | Implementing the Practices of Cooperative Organizational Culture by SLR | % of Practices Identified through SLR |
|---|---|---|
| 10. | A common set of development tools and policies should be used to facilitate a common understanding. | 26% |
| 11. | A communication protocol should be established so that it can overcome communication issues and barriers. | 24% |

*3.3. Face-to-Face Meetings*

Face-to-face meetings define communication with team members. The important points are to focus on daily face-to-face meetings, follow normal work schedules, require no overtime, have strong obligations regarding customers, and give full authority to customers [11,27,28]. In agile software development, managers are more close with and more frequently communicate with programmers to adapt to the user's needs iteratively until finished. This could be considered hard on the developers [29–31].

For implementing the SF "face-to-face meetings", our SLR study also found five practices. Table 5 illustrates that face-to-face meetings can be achieved by following five critical practices (% ≥ 20):

- Interactions among the developers should be closer and more frequent as compared to the traditional method of development.
- Regular face-to-face meetings should be established, as they provide clear communication.
- A proper working schedule should be followed.
- Customer commitment and presence, along with freedom, should be provided.
- Sharing problems and issues from their work can help the team to keep track of their progress.

**Table 5.** Practices for implementing face-to-face meetings.

| S. No. | Practices for Implementing Face-to-Face Meetings Identified through SLR | % of Practices Identified through SLR |
|---|---|---|
| 1. | Interaction among the developers is more close and frequent as compared to the traditional method of development. | 26% |
| 2. | Regular face-to-face meetings should be established as they follow the strong communication method which is very successful. | 28% |
| 3. | Proper working schedule should be followed. | 29% |
| 4. | Customer commitment and presence along with freedom should be provided. | 30% |
| 5. | Meeting with the managers should not send the wrong message that they are meeting because of any issue or missed deadline. | 17% |
| 6. | Sharing problems and issues from their work can help the team to keep track of their progress. | 20% |

*3.4. Dedicated Management*

Dedicated management standards prompt different scheduling issues—for example, accomplishing on-time conveyance, since it is difficult to appraise the correct volume of work required in the project arranging stage [32,33]. With respect to programming, this basic success factor (SF) is being considered in ongoing investigations [5,15,34].

We have found five practices for achieving "dedicated management" through SLR. From Table 6 we have noted that the most suitable four practices/solutions (% ≥ 20) for "dedicated management" are:

- Effective crew coordination is in large part a function of effective team communications.
- The communication elements required of a worker in a front line manufacturing team are the same as in a management team.
- Management concepts, emphasizing how to be an efficient team player.
- For better implementation and maintenance phases, they plan accordingly.

**Table 6.** Practices for implementing dedicated management.

| S. No. | Practices for Implementing Dedicated Management Identified through SLR | % of Practices Identified through SLR |
|---|---|---|
| 1. | State the effective crew coordination is in large part a function of effective Team communications. | 23% |
| 2. | The communication elements required of customer in a front line manufacturing team be the same as management team. | 20% |
| 3. | Managing efficient team properties like team buildup, meeting tasks deadlines, policy and context, life cycles, and needs of the management. | 17% |
| 4. | Management concepts, emphasizing that how to be an efficient team player. | 21% |
| 5. | For better implementation and maintenance phases they plan accordingly. | 23% |

*3.5. Team Competency—Agile Development Expertise*

Team competency according to agile development knowledge is to produce a product the customer approves of in good time. Software programming groups have to combine efficiency, flexibility, and first-class development practices with an eye to competing with the fast growing international world. In order to stay competitive within the world, software organizations observe agile improvement technique [35–37]. As with agile development, this technique focuses less on specifics, which help the development groups to experiment, permitting creativity to give them answers for detailed requirements [19,38,39].

For implementing SF "team competency—agile development expertise" our SLR study found five practices. Table 7 shows that "team competency—agile development expertise" can be best achieved by ensuring the following five practices (% ≥ 20):

- Remembering that agile development is being promoted as a means for reducing time, improving quality, increasing productivity, gaining efficiency, and becoming cost effective.
- Continuous feedback to the customer that supports successful development and delivery of the software; and continuous planning, integration, and testing.
- Integrate efficiency, flexibility, and quality into development practices in order to compete within the market.
- Communicate: the team in general deal with the various issues and problems that arise from the development efforts.
- Competency-based management has been a growing trend in organizations.

*3.6. Agile Development Environment*

There are many well-known agile methods (such as extreme programming (XP) and Scrum), and many native agile methods fall into this category [40–42]. Organizations have modified the existing agile methodologies to custom meet their development needs. Agile development teams demonstrate distributed control and flexible structures [43–45].

**Table 7.** Practices for implementing team competency—agile development expertise.

| S. No. | Practices for Implementing Team Competency—Agile Development Expertise Identified through SLR | % of Practices Identified through SLR |
|---|---|---|
| 1. | Agile development is being promoted as a mean for reducing time, improving quality, increasing productivity, gaining efficiency, and becoming cost effective | 21% |
| 2. | Continuous feedback to the customer that supports successful development and delivery of the software; and continuous planning, integration and testing. | 27% |
| 3. | Integrate efficiency, flexibility and quality in their development practices in order to compete with the fast development. | 31% |
| 4. | The team can assess its objectives, priorities, time management allocations and performance assessment. | 17% |
| 5. | Communicate, the team in general deal with the various issues and problems that arise from the development effort. | 20% |
| 6. | Competency-based management has been a growing trend in organizations. | 20% |

Table 8 shows six practices for SF "agile development environment". From Table 8 it is clear that the most suitable practices/solutions are the following (% ≥ 30):

- The use of Web applications and software for agile artifacts.
- Communication methods for better communication with remote teams.
- A hybrid approach should be adopted using both local and remote teams.

**Table 8.** Practices for implementing an agile development environment.

| S. No. | Practices for Implementing Team Competency—Agile Development Expertise Identified through SLR | % of Practices Identified through SLR |
|---|---|---|
| 1. | Distribute work equally and assign ownership to every individual. | 26% |
| 2. | Use of Web applications and software for agile artifacts. | 32% |
| 3. | Communication methods for better communication with remote teams. | 36% |
| 4. | Commitment should be based on product and delivery satisfaction. | 26% |
| 5. | Room for proper training along with the demos should be done frequently. | 23% |
| 6. | Hybrid approach should be adopted with both sets of team members from local and remote teams. | 31% |

*3.7. Team Encouragement*

All team members should always encourage their followers and escalate their influence in the group so as to develop a unified team and streamline the determination of the group towards gaining organizational achievements [46,47]. Team members can validate individual changes and assign the tasks in agreement with their strengths. The distinct contributions of team members should be respected and glorified in the team, which reinforces the group and makes members more unified and favorable toward work [48,49]. Given how software must integrate, good relationships are vital. Stimulating the use of similar reasonable practices is important, and encouragement helps with acceptance [47,50].

We have found seven practices for implementing "team encouragement" through SLR (Table 9). The most suitable five practices/solutions (% ≥ 25) for implementing ASDM at a large scale are:

- Team members support their followers and appreciate their contributions.
- Provisions of coaching and positive feedback should be given to team members for higher goal achievement.
- Gather performance feedback from team members.
- Assess and cultivate teamwork skills.
- Create a dialogue of feedback with each team member to encourage, challenge, and inform each other.

**Table 9.** Practices for implementing team encouragement.

| S. No. | Practices for Implementing Team Encouragement Identified through SLR | % of Practices Identified through SLR |
| --- | --- | --- |
| 1. | Team members support the followers and appreciate their contribution. | 28% |
| 2. | Provision of the coaching and positive feedback should be given to the performing and motivated team members for higher goal achievement. | 26% |
| 3. | Gather Performance Feedback from Team Members. | 26% |
| 4. | Assess and Cultivate Teamwork Skills. | 25% |
| 5. | Learn and acknowledge your employees' personal career objectives. | 20% |
| 6. | Show team respect through consistency and empowerment. | 23% |
| 7. | Create a dialogue of feedback with each team member to encourage, challenge, and inform each other. | 25% |

*3.8. Customer Satisfaction*

The main goal of this section is customer satisfaction, which helps the developers in meeting the ever changing customer requirements along with quick delivery of requirements and constant updates to their product [20,51].

ASD achieves customer satisfaction by providing valuable services early and consistently. Once large-scale and distributed agile methods are developed, the main focus remains the same. Especially providing customers with valuable solutions to achieve satisfaction, which can only be achieved by following the core values and guiding principles of the approach. As an example, defect count will be an enclosed method of package testing; however, at the same time the acceptable quantity of defects may be reduced by the client [25,46,52].

For implementing the "customer satisfaction" SF, our SLR study found five practices. Table 10 shows that customer satisfaction can be best achieved by following the five critical practices below (% ≥ 25):

- Perfect quality along with the right time, place, and price.
- The effective value of a critical measure defined by the customer is also created in the customer project.
- Agile methods work better with changing requirements.
- Software delivery does not depend on the availability of the customers on site always.
- Satisfying continuous customer requirements and being able to deliver continuously.

**Table 10.** Practices for implementing customer satisfaction.

| S. No. | Practices for Implementing Customer Satisfaction Identified through SLR | % of Practices Identified through SLR |
|:---:|---|:---:|
| 1. | Perfect Quality along with the Right Time, Place, and Price. | 26% |
| 2. | The effective value of a critical measure defined by the customer is also created in the customer project. | 25% |
| 3. | Agile methods works better with changing requirements. | 28% |
| 4. | Software delivery does not depend on the availability of the customers on site always. | 28% |
| 5. | Satisfying continuous customer requirements and being able to deliver continuously. | 29% |

*3.9. Strong Collaboration with the Customer*

The group with the highest maturity is "collaboration" and is assigned to the high maturity category. Due to the combination of practice and collaboration, communication, learning, and interaction, it is given this name—a completely different aspect of the team's working methods [26,53]. As organizations increasingly rely on agile software systems to develop and explore the principles of agile software system development methods, victimization theory will generate insight, but agile software system development methods provide value to software system development teams, such as interpreting each agile application. This helps improve the level of collaboration between software system development teams and customers [54–56].

We have found 12 practices for SF "strong collaboration with customer" through SLR study. From Table 11 we have noted that the most suitable twelve practices/solutions (% ≥ 25) for implementing strong collaboration with customer are:

- Utilize the time zone differences by managing the working hours between the two sites in such a way that can lead towards 24 h development.
- The 24 working hours should be divided between the time zones that development does not stop.
- Visits to the sites should be promoted.
- Enhance active involvement in all aspects of development for better understanding with customer.
- Confidence should be given to both formal and informal meetings among all stakeholders.
- Data should be shared with the customer at all stages.
- Software configuration management should be used to manage the different components of the software system.
- Increased dependency on the partners should be encouraged.
- Regular meetings with the customer will give better reviews and feedback
- Communication among team members should be ensured.
- Good management among one another, the project team and top management along with the customers.
- The role of the customer should be expanded to process of development of product along with discussing the important features, prioritizing requirements and user demands from time to time.
- Closely engage the customers into the development phase.

*3.10. Sustainable Planning*

Sustainable planning adoption is an attempt by software experts to solve current climate problems, and it can also improve the overall economic performances of software program organizations. Software program practitioners use sustainable practices to be

useful to society, climate, and humanity [57,58]. Existing techniques particularly stress realizing sustainability when software program experts improve software. The sustainability of software programming methods involves the growth, use, and processing of computers, servers, and related subsystems, and the effectiveness and efficiency of video display units, printers, storage devices, networks, and communication structures. However, there is often little or no impact on the surrounding environments [48,59].

**Table 11.** Practices for implementing strong collaboration with customer.

| S. No. | Practices for Implementing Strong Collaboration with Customer | % of Practices Identified through SLR |
|---|---|---|
| 1. | The 24 working hours should be divided between the time zones that development does not stop. | 30% |
| 2. | Visits to the sites should be promoted. | 28% |
| 3. | Enhance active involvement in all aspects of development for better understanding with customer. | 29% |
| 4. | Confidence should be given to both formal and informal meetings among all stakeholders. | 26% |
| 5. | Data should be shared with the customer at all stages. | 28% |
| 6. | Software configuration management should be used to manage the different components of the software system. | 28% |
| 7. | Increased dependency on the partners should be encouraged. | 29% |
| 8. | Regular meetings with the customer will give better reviews and feedback. | 28% |
| 9. | Communication among team members should be ensured. | 29% |
| 10. | Good management among one another, the project team and top management along with the customers. | 28% |
| 11. | The role of the customer should be expanded to process of development of product along with discussing the important features, prioritizing requirements and user demands from time to time. | 25% |
| 12. | Closely engage the customers into the development phase. | 25% |

For SF "sustainable planning" our SLR study found six practices. Table 12 shows that "sustainable planning" can be best achieved by keeping an eye on the following five practices (% ≥ 25):

- Attempt to understand and improve the current climatic and economic problem of software organizations.
- Current practices only focus on sustainability for software developers.
- Practices in software processes to efficiently ensure services, lower costs, and better systems control
- The deployment and implementation of software systems for better contribution to sustainable processes.
- The adoption of agile methodology for sustained software organizations varies with the network physical infrastructure.

**Table 12.** Practices for implementing sustainable planning.

| S. No. | Practices for Implementing Sustainable Planning Identified through SLR | % of Practices Identified through SLR |
|---|---|---|
| 1. | Attempt to understand and improve the current climatic and economic problem of software organizations. | 28% |
| 2. | Current practices only focus on sustainability for software developers. | 30% |
| 3. | Practices in software processes to efficiently ensure services, lower costs, and better systems control. | 34% |
| 4. | The deployment and implementation of software systems for better contribution to sustainable processes. | 25% |
| 5. | The adoption of agile methodology for sustained software organizations varies with the network physical infrastructure. | 25% |
| 6. | How to attain present goals without compromising the future generations and for them to achieve their goals. | 24% |

*3.11. Use of Automated Software Tools*

Since agile practices emphasize interactions between individuals and tools, and process and customer collaboration rather than contract negotiation (agile declarations), instinctively, the nature of the organization and its work is a key factor. As expected, there are tools developed by software engineering researchers to support development teams. Chau and Maurer [60] have shared a set of tools for knowledge sharing and inter-team coordination. Awareness provides automatic tools to improve focus to create habits that lead to less stress [5].

Table 13 represents nine practices for "use of automated software tools". From Table 13 it is clear that the most suitable practices/solutions found for "use of automated software tools" are the following six (% ≥ 25):

- Install disaster recovery software application.
- The reuse of software modules for knowledge using coding.
- Manage efficiently control life cycle to avoid data redundancy.
- Efficient coding applications for software development.
- Delete old and unused servers from the database.
- New software systems should operate in energy efficient methods.

**Table 13.** Practices for implementing use of automated software tools.

| S. No. | Practices for Implementing Use of Automated Software Tools Identified through SLR | % of Practices Identified through SLR |
|---|---|---|
| 1. | Knowledge management tools should be used to carry out data de-duplication. | 14% |
| 2. | Install disaster recovery software application. | 26% |
| 3. | Data de-duplication rightsizing software equipment, storage tiring. | 19% |
| 4. | The reuse of software modules for knowledge using coding. | 28% |
| 5. | Manage efficiently control life cycle to avoid data redundancy. | 26% |
| 6. | Efficient coding applications for software development. | 41% |
| 7. | Delete old and unused servers from the database. | 28% |
| 8. | New software systems should operate in energy efficient methods. | 31% |
| 9. | Understand the effect of software usage on the Environment. | 23% |

### 3.12. Scheduled Trainings for Team Members

The arrangement of trainings depends upon the experience of the staff or observations of the team members, and the need is dependent upon the number of successful members, along with communication and collaboration. An important matter of interest in this scenario deals with the measurement of the percentage of workers which is essential for the victorious execution of agile methods. Agile relies on an "uninterrupted process of mastering skills", which is the main dissimilarity between "agile" and other methods [37,61].

Table 14 illustrates that for the SF "training for the team we can get better results faster", five practices are the products of our SLR. To achieve SF "scheduled training for team members", we have to adopt the following two practices as the threshold was decided to be (% ≥ 20):

- Individual competency should be streamlined with the team requirements for performance and efficiency.
- By providing training for the team, we can get better results faster.

**Table 14.** Practices for implementing scheduled training for team members.

| S. No. | Practices for Implementing Scheduled Training for Team Members Identified through SLR | % of Practices Identified through SLR |
|---|---|---|
| 1. | Sharing information continuously will increase the agile practices. | 14% |
| 2. | Reduced delivery schedules and increased return on investments. | 19% |
| 3. | Individual competency should be streamed with the team requirements for it to perform better and efficiently. | 21% |
| 4. | Arrange Trainings and meeting for the team to get work together effectively and increase competency levels. | 19% |
| 5. | Training for the team we can get better results faster. | 23% |

### 3.13. Strong Collaboration and Communication

In order to have fruitful alliance, team participants are required to have a teamwork to figure out "how the collective work is carried out on remote sites", "how to ponder upon different associated matters", "how to interact with each other for better results", and last but not least, "how to tackle the different problems which arise at different points and during efforts", how to figure out the solutions for those. A smooth way to settle the mentioned factors is required, which should be accompanied with strong collaboration and communication [42]. An issue which needs to be tackled and which is one of the important is "to have communication in order to work over same characteristic but at different size by large teams".

Table 15 represents five practices for "strong collaboration and communication". From Table 15 it is clear that the most suitable practice/solution found for "strong collaboration and communication" is the following (% ≥ 20):

- Frequent visits between team members and customers to maintain collaboration.

### 3.14. Risk Management

Risk management in ASDM reveals how security features may be organized into agile software development methods while using a large-scale developmental team. Proper planning at the requirement engineering phase may positively manage the risks in projects [9]. The simplest way to manage the risks is to categorize the risks into high and low criticality [46].

**Table 15.** Practices for implementing strong collaborations and communications.

| S. No. | Practices for Implementing Strong Collaborations and Communications Identified through SLR | % of Practices Identified through SLR |
|--------|---------------------------------------------------------------------------|---------------------|
| 1. | Remote communication with their team to work effectively and efficiently. | 15% |
| 2. | Frequent visits between team members and customers to maintain collaboration. | 23% |
| 3. | Focus on risks related to customers, communication and stakeholders. | 19% |
| 4. | Deliver final software to the end users more quicker than traditional approaches. | 17% |
| 5. | Involve the customer in agile projects. | 14% |

Table 16 represents eight practices for "risk management". From Table 16 it is clear that the most suitable practices/solutions found for "risk management" were the following (% ≥ 20):

- Extensive work has been done in Risk Management and present work that is needed for further consideration.
- Customer commitments and presence have full authority over other issues.
- Study the current approaches and find out new ways of new areas to explore.
- If risk is planned carefully initially the percentage of failure is reduced.
- If risk is managed carefully the chances of failure are reduced.
- Risks associated with high company effect are identified by the departments.
- Assessed, and mitigated the risk properly and communicate to your entire organization.

**Table 16.** Practices for implementing risk management.

| S. No. | Practices for Implementing Risk Management Identified through SLR | % of Practices Identified through SLR |
|--------|---------------------------------------------------------------|---------------------|
| 1. | Extensive work has been done in Risk Management and present work that is needed for further consideration. | 26% |
| 2. | Customer commitments and presence have full authority over other issues. | 23% |
| 3. | Study the current approaches and find out new ways of new areas to explore. | 25% |
| 4. | If risk is planned carefully initially the percentage of failure is reduced. | 28% |
| 5. | If risk is managed carefully the chances of failure are reduced. | 21% |
| 6. | Risks associated with high company effect are identified by the departments. | 23% |
| 7. | Assessed and mitigated the risk properly and communicate to your entire organization. | 24% |
| 8. | After you have assumed the initial risk assessment then you should put the relevant risk controls for mitigation and monitoring. | 14% |

### 3.15. Knowledge Sharing Management

Via constructive and efficacious information distribution between different teams, a model will not be built up without proper planning or without well-defined and well identified structure, or with no awareness of the difficulties which may arise due to changes. Methodical reciprocity and information distributing strategies between team members and team leaders is mandatory for successful outcomes [41,62].

Table 17 represents seven practices for "knowledge sharing management". From Table 17 it is clear that the most suitable practices/solutions found for "knowledge sharing management" are the following (% ≥ 25):

- Use latest technology and processes for knowledge sharing and management.
- Establish how new procedures can be introduced into the workforce for smooth information sharing among new and old team employees.

- Team level collaboration and knowledge should be shared for adaptability for both large and small scale teams.

**Table 17.** Practices for implementing knowledge sharing management.

| S. No. | Practices for Implementing Risk Management Identified through SLR | % of Practices Identified through SLR |
|---|---|---|
| 1. | Sharing of knowledge, expectations and concerns in work ethics should be done through video conferences, emails, and calls. | 24% |
| 2. | Use latest technology and processes for knowledge sharing and management. | 26% |
| 3. | Variance analysis should be used when and where it is needed. | 20% |
| 4. | Establish how new procedures can be introduced into the workforce for smooth information sharing among new and old team employees. | 26% |
| 5. | Convert tacit knowledge to explicit knowledge by documentation and process description. | 20% |
| 6. | Implement new domain and technical trainings to update the database and better profiling of the employees. | 24% |
| 7. | Team level collaboration and knowledge should be shared for adoptability for both large and small scale teams. | 26% |

### 3.16. Quality Production Using Pair Programming

It is widely used in various agile software development methods, and is recommended for various agile software development methods, including function-driven development, Scrum, lean software development, and Crystal and dynamic system development methods [42]. It is one of the notorious aspects of many agile system development methods (especially XP). Pair programming is the basic practice of XP. In local programming, traditionally, a programmer is responsible for installing and testing their own code. In pair programming, each code component is generated by a group of programmers at the same workstation. The pair consists of two roles, a driver that controls the mouse, a keyboard, or other input device to write code tests; and a navigator to comply with quality assurance, ask questions, consider alternatives, and point out defects. These are considered equal, and roles and partners are exchanged over time [52].

Table 18 represents 13 practices for "quality production using pair programming". From Table 18 it is clear that the most suitable practices/solutions found for "quality production using pair programming" are the following (% ≥ 20):

- Better competition through service provision.
- Provision of best services design and execution.
- Follow strict time development schedule.
- Establish mutual trust.
- Standard RE models should be used for conducting Requirements Engineering phase.
- Improve client-vendor communication.
- Ensure that the client requirements of response time, flexibility, usability and reliability are met.

### 3.17. Mechanism for Change Management

A mechanism for change management that is extremely inflexible and strict further worsens the problems of poor responsiveness to customers and market demand. Quality costs are increasing and growing because of a long-term testing and repair period after already extensive investment. Management believes that the inspiration of the staff is low because people are mainly engaged in small silos of professional software engineering with limited product knowledge [39].

**Table 18.** Practices for implementing quality production using pair programming.

| S. No. | Practices for Implementing Quality Production Using Pair Programming Identified through SLR | % of Practices Identified through SLR |
|--------|------|------|
| 1. | Improved capability of the various vendors of better implementation of SPI certifications, i.e., CMM and CMMI. | 15% |
| 2. | Improving product quality through proper monitoring. | 17% |
| 3. | Better competition through service provision. | 20% |
| 4. | Better ways of providing interaction among team players for better tacit knowledge sharing. | 14% |
| 5. | Hiring process should be based on good job skills. | 15% |
| 6. | Provision of best services design and execution. | 24% |
| 7. | Follow strict time development schedule. | 28% |
| 8. | Establish mutual trust. | 24% |
| 9. | Offer quality management trainings. | 19% |
| 10. | Standard RE models should be used for conducting Requirements Engineering phase. | 23% |
| 11. | Improve client-vendor communication. | 28% |
| 12. | Ensure that the client requirements of response time, flexibility, usability and reliability are met. | 20% |
| 13. | Ensuring that there is provision of global talent and also the delivery models should be world class. | 19% |

Table 19 represents eight practices for "mechanism for change management". From Table 19 it is clear that the most suitable practices/solutions found for "mechanism for change management" are the following five practices (% ≥ 20):

- In different situations the capacity should work effectively, as change management is known to be unpredictable.
- Change in the organization needs to be done in a collaborative way with the old problems being shared with the employees along with the new benefits of the new system.
- Frequently meet with your change team employees and encouraging them to share their feedback and provide what works and what is not working.
- Awareness to the employees that this is a learning curve and it will have its share of questions, concerns and suggestions which are all entertained.

**Table 19.** Practices for implementing mechanism for change management.

| S. No. | Practices for Implementing Mechanism for Change Management Identified through SLR. | % of Practices Identified through SLR |
|--------|------|------|
| 1. | Efficient problem solving, better communication and strong collaboration among team members. | 15% |
| 2. | In different situations the capacity should work effectively, as change management is known to be unpredictable. | 20% |
| 3. | Team members should have clear set priorities and they should be kept moving ahead to reach other goals. | 25% |

**Table 19.** *Cont.*

| S. No. | Practices for Implementing Mechanism for Change Management Identified through SLR. | % of Practices Identified through SLR |
|--------|----------------------------------------------------------------------------------------------------------------------|---------------------------------------|
| 4. | Change in the organization needs to be done in a collaborative way with the old problems being shared with the employees along with the new benefits of the new system. | 23% |
| 5. | Sudden changes to the working environment should not be introduced as it is rejected and employees are not able to adjust to it. | 12% |
| 6. | Release of information change should be done quickly and then the incremental steps involved should be performed. | 17% |
| 7. | Frequently meet with your change team employees and encouraging them to share their feedback and provide what works and what is not working. | 20% |
| 8. | Awareness to the employees that this is a learning curve and it will have its share of questions, concerns and suggestions which are all entertained. | 23% |

*3.18. Leadership Strong Commitment and Team Autonomy*

Agile projects tend to have a combination of strong senior level management, and have strong obligations. For a successful agile software developmental project, the factors of quality, time, and scope are very important and cannot be neglected for aa project of any size [19].

Table 20 represents six practices for "leadership strong commitment and team autonomy". From Table 20 it is clear that the most suitable practices/solutions found for "leadership strong commitment and team autonomy" are the following five practices (% ≥ 20):

- Fostering trustful ties with the client.
- Additional favors in development of mutually beneficial partnership.
- Apply effort and other resources for maintaining on going relationships.
- The other factors related to customer satisfaction and service along with both financial performance, internal business and growth should be considered.
- Better learning requires to understand and listen to other ideas and opinions.

**Table 20.** Practices for implementing leadership strong commitment and team autonomy.

| S. No. | Practices for Implementing Leadership Strong Commitment and Team Autonomy Identified through SLR | % of Practices Identified through SLR |
|--------|----------------------------------------------------------------------------------------------------|---------------------------------------|
| 1. | Fostering trustful ties with the client. | 24% |
| 2. | Additional favors in development of mutually beneficial partnership. | 20% |
| 3. | Apply effort and other resources for maintaining on going relationships. | 26% |
| 4. | "Future orientation" plans both long range and short range should be discussed. | 18% |
| 5. | The other factors related to customer satisfaction and service along with both financial performance, internal business and growth should be considered. | 24% |
| 6. | Better learning requires to understand and listen to other ideas and opinions. | 23% |

*3.19. Pilot Project in Case of No Experience*

Pilot projects, pilot studies, and pilot tests are small-scale preliminary studies designed to assess feasibility, time, cost, and adverse events; and to improve research or project planning prior to conducting a comprehensive research project. Therefore, the pilot project may not be suitable for case studies [5]. The system will identify individual and team contributions and reward the results of the agile pilot project. The organization has a reward

system for agile behavior. The project team is co-located, meaning that all team members work in similar areas to facilitate communication and easy and stable communication [47].

Table 21 represents six practices for "pilot project in case of no experience". From Table 21 it is clear that the most suitable practices/solutions found for "pilot project in case of no experience" are the following five practices (% ≥ 20):

- Confidence in the working principle of Agile would be so suitable and also gaining acceptance of its approach is increased.
- Various learning experiments vied insights into the problems and who they were mitigated.
- To have better communication and contact constant contact was maintained in a working environment.
- Pilot projects success and failures tend to give stakeholders information at an early stage about the project.
- If the pilot project was not received well but over all the idea is good and should be pushed, gives the project team staff an opportunity to adopt a new strategy for better success.

**Table 21.** Practices for implementing pilot project in case of no experience.

| S. No. | Practices for Implementing Pilot Project in Case of No Experience Identified through SLR | % of Practices Identified through SLR |
|---|---|---|
| 1. | Confidence in the working principle of Agile would be so suitable and also gaining acceptance of its approach is increased. | 20% |
| 2. | Various learning experiments vided insights into the problems and who they were mitigated. | 23% |
| 3. | To have better communication and contact constant contact was maintained in a working environment. | 26% |
| 4. | By using and creating project through Agile more confidence was given to the development team members. | 23% |
| 5. | Pilot projects success and failures tend to give stakeholders information at an early stage about the project. | 24% |
| 6. | If the pilot project was not received well but over all the idea is good and should be pushed, gives the project team staff an opportunity to adopt a new strategy for better success. | 14% |

*3.20. Training and Learning and Briefing of Top Management on Agile*

Leadership skills are an important factor of an individual and create a fruitful team. Leadership skills create a major impact on organization management and the culture of an organization; demonstrative leadership skills are vital for the achievement with agile methods. Governance, team orientation, idleness, learning, and sovereignty need cumulative addressing in light of one another's beliefs. Leadership is an collective ideology of agile teams and can be hard to do well [25].

Table 22 represents three practices for "training and learning and briefing of top management on agile". From Table 22, it is clear that the most suitable practices/solutions found for "training and learning and briefing of top management on agile" were the following three practices (% ≥ 20):

- Agile software development has specific management roles which keep a hold on both the unique and difficult processes.
- Following the roles and techniques for better planning and flexibility and learning.
- Have a better organization and learning mechanism to deal with the complexity and unpredictability of agile software projects.

**Table 22.** Practices for implementing training and learning and briefing of top management on agile.

| S. No. | Practices for Implementing Training and Learning and Briefing of Top Management on Agile Identified through SLR | % of Practices Identified through SLR |
|---|---|---|
| 1. | Agile software development has specific management roles which keep a hold on both the unique and difficult processes. | 25% |
| 2. | Following roles and techniques for better planning and flexability and learning. | 23% |
| 3. | Have a better organization and learning mechanism to deal with the complexity and unpredictability of agile software projects. | 28% |
| 4. | To better understand the specific management functions a unique and complex process and its activities should be implemented. | 14% |
| 5. | Training and learning approached should be increased in terms of skills diversity. | 13% |
| 6. | National Training Framework should be adopted which is linked to accredited training. | 15% |

*3.21. Requirement Management Using Agile-Oriented Requirement Management Process*

Many traditional project teams will have trouble trying to define all requirements in advance, which is usually misleading because developers will actually read and follow what is contained in the requirements document. The reality is that the requirements documentation is usually not enough. No matter how much effort is invested, the requirements will still change. In the end, developers will eventually directly request information from their stakeholders.

Table 23 represents four practices for "requirements management using agile-oriented requirement management process". From the Table 23, it is clear that the most suitable practices/solutions found for "requirements management using agile-oriented requirement management process" are the following practices (% ≥ 15):

- They turn the business potential into actual competitiveness on the market.
- Requirement management is all about learning and documenting the work to be performed by the project, and ensuring compatibility with resources.
- Resolving, analyze, specify, validate, and manage software requirements.
- Dealing with crosscutting requirements systematically, and this can be integrated by agile software development methodologies.

**Table 23.** Practices for implementing requirement management using agile-oriented requirement management process. Here, the threshold is 15%.

| S. No. | Practices for Implementing Requirements Management Using Agile-Oriented Requirement Management Process Identified through SLR | % of Practices Identified through SLR |
|---|---|---|
| 1. | Requirements management is all about learning and documenting the work to be performed by the project, and ensuring compatibility with resources. | 15% |
| 2. | Resolving, analyze, specify, validate and manage software requirements. | 17% |
| 3. | They turn the business potential into actual competitiveness on the market. | 20% |
| 4. | They define the requirements in terms of goals which are well understood by the stakeholders. | 14% |
| 5. | Deals with crosscutting requirements systematically and this can be integrated by agile software development methodologies. | 15% |

## 4. Study Limitations

In this section, the threats to validity concerning the SLR study are discussed. By using an SLR procedure, we mined practices/solutions for each of the SFs concerning the adoption of ASDM at a large scale from management perspectives, but how valid are our findings? Related to internal validity, studies have not explicitly mentioned their reasons for reporting solutions of the ASDM transformation factors. We are unable to control this threat. However, we have tried our level best by validating the results through interrelated test, i.e., checking the results by all authors and deciding whether to include the practices or exclude them. This detailed checking was time consuming, but it gave us assurance. Furthermore, one possible threat to internal validity is that for any practice/solution in an article, the author may not in fact have described the underlying causes of practices for SF implementations. Concerning the threat of external validity, our sample was composed of the articles reporting data from diverse countries. We have full confidence in our results because we found more similarities than differences in the outcomes. This provides evidence for generalization. We have conducted our SLR as part of a collaboration and consulted the software engineering research group at the University of Engineering and Technology, Mardan (SERG_UETM) for validation of the search strings. With the increasing number of papers in ASDM, our SLR process may have missed out some relevant papers. However, as in other SLRs, this was not a systematic omission [17]. To deal with subjectivity and researcher biases, we also performed an inter-rater reliability check on every step of the SLR. We do not claim that we have included all digital libraries, so when executing our SLR process it was possible to miss some relevant papers. The first reason for that was the abundance of papers on ASDM. The second reason was that we did not have access to every digital library due to a lack of resources. However, the included digital libraries were more than enough for the synthesis of results in our study. According to other academic investigators, such as [17,62], when using SLR as a method for data collection, this is not a methodical omission.

## 5. Conclusions and Future Work

Initially we identified 147 practices, in total, concerning SFs important for the adoption of ASDM at large scales from a management perspective. After applying the threshold, the number was reduced to 99. Our results revealed that focusing on these practices/solutions can help software project managers to transform from conventional software development methods to scaled agile software development methods. Regardless of all the stated limitations, we are confident in that our study will contribute to academia and the industrial domain. This study will:

- Provide software project managers knowledge that can assist them in implementing and adopting ASDM at large scales. Our results recommend that software project managers should adopt all of the reported practices for SFs, especially those reported with greater percentages.
- Increase team cohesiveness, as it will guide both sides toward understanding each other's requirements and goals, in order to sustain long term commitment.
- Provide assistance of understanding SF practices to ensure a successful transformation.

We have noted the following points, as a future plan, from the findings of this study:

- We will validate the practices identified through the SLR by conducting an empirical investigation of the agile software development industry.
- The practices/solutions in scaled ASDM from the team's perspective will be identified and analyzed.
- We will analyze the critical risk in the transformation to ASDM from a management perspective.
- We shall determine the underlying reasons for why some factors are not important for specific groups of ASD companies.

Our future work will focus on the development of a scaling agile adoption assessment maturity model (SAAAMM). This paper provides input for the development of the second phase of the SAAAMM, such as the identification of various practices of SFs. The SAAAMM will assist software project managers in the transformation to ASDM. The SAAAMM will provide guidance and boost the work that has been undertaken on the development of frameworks and models for ASDM transformation.

**Funding:** The National Research Foundation of Korea (NRF), funded by the Ministry of Education under grant NRF-2018R1A6A1A03025109; and in part by the National Research Foundation of Korea (NRF), funded by the Korean Government (MSIT) under grant NRF-2019R1A2C1006249.

**Acknowledgments:** This work was supported in part by the Basic Science Research Program through the National Research Foundation of Korea (NRF), funded by the Ministry of Education under grant NRF-2018R1A6A1A03025109; and in part by the National Research Foundation of Korea (NRF), funded by the Korean Government (MSIT) under grant NRF-2019R1A2C1006249.

**Conflicts of Interest:** The authors declare no conflict of interest.

## Appendix A

| S. No. | Paper Title | Authors/Date of Review |
|---|---|---|
| 1 | A survey study of critical success factors in agile software projects | Chow, T., & Cao, D. B. (2008). |
| 2 | Communities of Practice in a Large Distributed Agile Software Development Organization — Case Ericsson | Paasivaara, M., & Lassenius, C. (2014). |
| 3 | Risks in distributed agile development: A review | Shrivastava, S. V., & Rathod, U. (2014) |
| 4 | The impact of inadequate customer collaboration on self-organizing Agile teams | Hoda, R., Noble, J., & Marshall, S. (2011). |
| 5 | A Framework for Understanding the Factors Influencing Pair Programming Success | Ally, M., Darroch, F., & Toleman, M. (June 2005). |
| 6 | A Model for Adopting Sustainable Practices in Software Based Organizations | Mazlina, A. M., & Awanis, R. (2017). |
| 7 | A Quantitative Study on Critical Success Factors in Agile Software Development Projects; Case Study IT Company | Nasehi, A. (2013). |
| 8 | A survey study of critical success factors in agile software projects | Chow, T., & Cao, D. B. (2008) |
| 9 | A survey study of critical success factors in agile software projects in former Yugoslavia IT companies | Stankovic, D., Nikolic, V., Djordjevic, M., & Cao, D. B. (2013). |
| 10 | Adopting Agile Software Development Practices: Success Factors, Changes Required, and Challenges | Misra, S. C. (2007). |
| 11 | Agile Adoption in IT Organizations | Ghani, I., & Bello, M. (2015). |
| 12 | Agile software development: adaptive systems principles and best practices | Meso, P., & Jain, R. (2006). |
| 13 | Challenges of Migrating to Agile Methodologies | Nerur, S., Mahapatra, R., & Mangalaraj, G. (2005). |
| 14 | Agile software development | Cockburn, A. (2002). |
| 15 | Agile principles and achievement of success in software development: A quantitative study in Brazilian organizations | de Souza Bermejo, P. H., Zambalde, A. L., Tonelli, A. O., Souza, S. A., Zuppo, L. A., & Rosa, P. L. (2014). |
| 16 | Agile process in software engineering and extreme programming (2009) | Abrahamsson, P., Marchesi, M., & Maurer, F. (2009). |
| 17 | Agile process in software engineering and extreme programming (2017) | Baskerville, P. A. R., Fitzgerald, K. C. B., & Wang, L. M. X. |
| 18 | Agile Processes in Software Engineering and Extreme Programming (2014) | Abrahamsson, P., Marchesi, M., & Maurer, F. (2009). |
| 19 | Agile Project Management: | Hoda, R., Noble, J., & Marshall, S. (2008) |

| S. No. | Paper Title | Authors/Date of Review |
|---|---|---|
| 20 | Agile Software Development Methodologies and Practices | Williams, L. (2010). |
| 21 | Agile transition and adoption human-related challenges and issues: A Grounded Theory approach | Gandomani, T. J., & Nafchi, M. Z. (2016). |
| 22 | Agile User Experience Development in a Large Software Organization: Good Expertise but Limited Impact | Kuusinen, K., Mikkonen, T., & Pakarinen, S. (2012) |
| 23 | The Top 10 Burning Research Questions from Practitioners | Freudenberg, S., & Sharp, H. (2010). |
| 24 | Are you biting off more than you can chew? A case study on causes and effects of over-scoping in large-scale software engineering | Bjarnason, E., Wnuk, K., & Regnell, B. (2012). |
| 25 | Challenges and success factors for large-scale agile transformations: A systematic literature review | Dikert, K., Paasivaara, M., & Lassenius, C. (2016). |
| 26 | Agile adoption story from NHN | Kim, E., & Ryoo, S. (July 2012). |
| 27 | Agile Success Factors A qualitative study about what makes agile projects successful | Kropp, M. (2015). |
| 28 | Cost and time project management success factors for information systems development projects | Sanchez, O. P., & Terlizzi, M. A. (2017). |
| 29 | Critical success factors for implementation of supply chain management in Indian small and medium enterprises and their impact on performance | Kumar, R., Singh, R. K., & Shankar, R. (2015). |
| 30 | Critical Success Factors for Rapid, Innovative Solutions | Lane, J. A., Boehm, B., Bolas, M., Madni, A., & Turner, R. (July 2010). |
| 31 | Identifying some important success factors in adopting agile software development practices | Subhas Chandra Misraa, Vinod Kumar, Uma Kumar 2009 |
| 32 | The impact of supply chain management practices on performance of SMEs | Lenny Koh, S. C., Demirbag, M., Bayraktar, E., Tatoglu, E., & Zaim, S. (2007). |
| 33 | Agile Practices for the Global Teaming Model | |
| 34 | The effects of internal resources and partnership quality on firm performance: An examination of Indian BPO providers | Lahiri, S., & Kedia, B. L. (2009). |
| 35 | Practices for Implementation of the Critical Success Factors in Software Outsourcing Partnership from Vendors' Perspective: A Literature Review | Ali, S., & Khan, S. U. (2016). |
| 36 | Research Review on Open Innovation: Literature Review and Best Practices | West, J., & Bogers, M. (2014). |
| 37 | Routine inter-dependencies as a source of stability and flexibility. A study of agile software development teams | Dönmez, D., Grote, G., & Brusoni, S. (2016). |
| 38 | Scaling up the Planning Game: Collaboration Challenges in Large-Scale Agile Product Development | Evbota, F., Knauss, E., & Sandberg, A. (May 2016). |
| 39 | Software teams and their knowledge networks in large-scale software development | Šmite, D., Moe, N. B., Š?blis, A., & Wohlin, C. (2017). |
| 40 | Strategic factors in agile software development method adaptation | Lal, R. (2011). |
| 41 | Success Factors for Building and Managing High Performance Agile Software Development Teams | Nguyen, D. S. (2016). |
| 42 | Success Factors of Agile Software Development | Misra, S. C., Kumar, V., & Kumar, U. (2006). |
| 43 | Success factors for implementation of novel decentralized diagnostics: How publicly funded multidisciplinary innovation networks can disrupt German Healthcare | Hanke, M. (2017). |
| 44 | Why Do Customer Relationship Management Applications Affect Customer Satisfaction? | Mithas, S., Krishnan, M. S., & Fornell, C. (2005). |
| 45 | The impact of supply chain management practices on performance of SMEs performance of SMEs | Lenny Koh, S. C., Demirbag, M., Bayraktar, E., Tatoglu, E., & Zaim, S. (2007). |
| 46 | Do Agile Software Development Practices Increase Customer Satisfaction in Systems Engineering Projects? | Kohlbacher, M., Stelzmann, E., & Maierhofer, S. (April 2011). |

| S. No. | Paper Title | Authors/Date of Review |
|---|---|---|
| 47 | Do not Forget to Breathe: A Controlled Trial of Mindfulness Practices in Agile Project Teams | den Heijer, P., Koole, W., & Stettina, C. J. (May 2017). |
| 48 | Educating for Sustainability: Competencies & Practices for Transformative Action | Frisk, E., & Larson, K. L. (2011). |
| 49 | Empirical studies of geographically distributed agile development communication challenges: A systematic review | Alzoubi, Y. I., Gill, A. Q., & Al-Ani, A. (2016). |
| 50 | Empirical Investigation on Success Factors in Adapting Agile Methodology in Software Development at Public Organizations | Alzoubi, Y. I., Gill, A. Q., & Al-Ani, A. (2016). |
| 51 | Excessive software development: Practices and penalties | Shmueli, O., & Ronen, B. (2017). |
| 52 | Exploring differences between smaller and large organizations' corporate governance of information technology | Wilkin, C. L., Couchman, P. K., Sohal, A., & Zutshi, A. (2016). |
| 53 | Exploring the role of lean thinking in sustainable business practice: A systematic literature review | Caldera, H. T. S., Desha, C., & Dawes, L. (2017). |
| 54 | Extreme programing and agile process in software engineering | Marchesi, M., & Succi, G. (2003). |
| 55 | Factors associated with the software development agility of successful projects | Sheffield, J., & Lemétayer, J. (2013) |
| 56 | Factors that motivate software engineering teams: A four country empirical study | Verner, J. M., Babar, M. A., Cerpa, N., Hall, T., & Beecham, S. (2014). |
| 57 | Practices for Software Integration Success Factors in GSD Environment | I lyas, M., & Khan, S. U. (June 2016). |
| 58 | From Anarchy to Sustainable Development: Scrum in Less Than Ideal Conditions | Therrien, I., & LeBel, E. (August 2009). |
| 59 | How agile are industrial software development practices? | Hansson, C., Dittrich, Y., Gustafsson, B., & Zarnak, S. (2006). |
| 60 | Identify and Classify Critical Success Factor of Agile Software Development Methodology Using Mind Map | El Hameed, T. A., Latif, M. A. E., & Kholief, S. (2016). |
| 61 | Implementing Agile project methods in globally distributed teams | Gillo Nilsson, C., & Karlsson, D. (2015). |
| 62 | Knowledge Management in New Product Teams: Practices and Outcomes | Gillo Nilsson, C., & Karlsson, D. (2015). |
| 63 | Knowledge Management: Practices and Challenges | |
| 64 | Knowledge Sharing: Agile Methods vs. Tayloristic Methods | Chau, T., Maurer, F., & Melnik, G. (2003, June). |
| 65 | Scale and responsive in large scale software development | Olsson, H., Sandberg, A., Bosch, J., & Alahyari, H. (2014). |
| 66 | Managerial Execution in Public Administration: Practices of Managers When Implementing Strategic Objectives | Sabourin, V., & Sefa, E. (2012). |
| 67 | Modeling continuous integration practice differences in industry software development | Ståhl, D., & Bosch, J. (2014). |
| 68 | Motivations and measurements in an agile case study | Layman, L., Williams, L., & Cunningham, L. (November 2004). |
| 69 | Moving from traditional to agile software development methodologies also on large, distributed projects. | Papadopoulos, G. (2015). |
| 70 | Operation of management control practices as a package—A case study on control system variety in a growth firm context | Sandelin, M. (2008). |
| 71 | Business Governance Best Practices of Virtual Project Teams | Hamersly, W. J. (2015). |
| 72 | Deepening Our Understanding of Communities of Practice in Large-Scale Agile Development | Paasivaara, M., & Lassenius, C. (July 2014). |
| 73 | process industry practices (pip): creation of industry practices | Mohla, D. L., Shannon, S. W., Sims, L., & Zerda, R. L. (September 1998). |
| 74 | Product focus software process improvement | Zahran, S. (2009). |
| 75 | Refining a model for sustained usage of agile methodologies | Senapathi, M., & Drury-Grogan, M. L. (2017). |
| 76 | Requirements for a successful buyer-supplier collaboration in new product development | ENGLESSON, P., & OHLIN, M. (2017). |

| S. No. | Paper Title | Authors/Date of Review |
|---|---|---|
| 77 | Tailoring Agile in the Large: Experience and Reflections from a Large-Scale Agile Software Development Project | Rolland, K. H., Mikkelsen, V., & Næss, A. (May 2016). |
| 78 | doing competencies well: best practices in competency modeling | Campion, M. A., Fink, A. A., Ruggeberg, B. J., Carr, L., Phillips, G. M., & Odman, R. B. (2011). |
| 79 | Factors that motivate software engineering teams: A four country empirical study | Verner, J. M., Babar, M. A., Cerpa, N., Hall, T., & Beecham, S. (2014). |
| 80 | Workplace design, complementarities among work practices and the formation of competencies. Evidence form Italian employees | Leoni, R. (2012). |
| 81 | Towards an Agile Feature Composition for a Large Scale Software Product Lines | Dehmouch, I. (May 2014). |
| 82 | Critical factors in software outsourcing: a pilot study | Oza, N., Hall, T., Rainer, A., & Grey, S. (2004, November). |

## Appendix B. List of Practices

1. Manage the product and separate quality assurance groups.

2. For any successful and effective team, the members need to be adaptable to any change in the task environment.

3. The ability to easily respond to change requirements of the customer.

4. The team members must be support each other for executing workflow of the project.

5. The members of the agile team should be mature in trust building to make it function

6. Briefly move the selected programmers to the Clients site.

7. Differences among other departments of the organizations should be reduced.

8. Focus on the importance of the Culture and understand what main blockage in the successful migration to the Agile development on a large scale.

9. Mutual understanding can be achieved by having active cross cultural communication programs through the use of short visits and face-to-face communication.

10. Different skill training which can be a combination of formal communication languages, client specific requirements, or either domain specific along with logical thinking.

11. There will be difference in Norms and Values from other cultures.

12. Move to the client's site for better communication.

13. Better use of middlemen can be made to adjust the communication barrier among clients.

14. Time management techniques should be used to understand the different time zones.

15. A common set of development tools and policies should be used to facilitate a common understanding.

16. A communication protocol should be established so that it can overcome communication issues and barriers.

17. Interaction among the developers is closer and more frequent as compared to the traditional method of development.

18. Regular face-to-face meetings should be established as they follow the strong communication method which is very successful.

19. Proper working schedule should be followed.

20. Customer commitment and presence along with freedom should be provided.

21. Meeting with the managers should not send the wrong message that they are meeting because of any issue or missed deadline.

22. Sharing problems and issues from their work can help the team to keep track of their progress.

23. State the effective crew coordination is in large part a function of effective Team communications.

24. The communication elements required of customer in a front-line manufacturing team be the same as management team.

25. Managing efficient team properties like team buildup, meeting tasks deadlines, policy and context, life cycles, and needs of the management.

26. Management concepts, emphasizing that how to be an efficient team player.

27. For better implementation and maintenance phases he plans accordingly.

28. Agile development is being promoted as a mean for reducing time, improving quality, increasing productivity, gaining efficiency, and becoming cost effective.

29. Continuous feedback to the customer that supports successful development and delivery of the software, and continuous planning, integration, and testing.

30. Integrate efficiency, flexibility, and quality in their development practices in order to compete with the fast development.

31. The team can assess its objectives, priorities, time management allocations and performance assessment.

32. Communicate, the team in general deal with the various issues and problems that arise from the development effort.

33. Competency-based management has been a growing trend in organizations.

34. Distribute work equally and assign ownership to every individual.

35. Use of Web applications and software for agile artifacts.

36. Communication methods for better communication with remote teams.

37. Commitment should be based on product and delivery satisfaction.

38. Room for proper training along with the demos should be done frequently.

39. Hybrid approach should be adopted with both sets of team members from local and remote teams.

40. Team members support the followers and appreciate their contribution.

41. Provision of the coaching and positive feedback should be given to the performing and motivated team members for higher goal achievement.

42. Gather Performance Feedback from Team Members.

43. Assess and Cultivate Teamwork Skills.

44. Learn and acknowledge your employees' personal career objectives.

45. Show team respect through consistency and empowerment.

46. Create a dialogue of feedback with each team member to encourage, challenge, and inform each other.

47. Perfect Quality along with the Right Time, Place, and Price.

48. The effective value of a critical measure defined by the customer is also created in the customer project.

49. Agile methods work better with changing requirements.

50. Software delivery does not depend on the availability of the customers on site always.

51. Satisfying continuous customer requirements and being able to deliver continuously.

52. The 24 working hours should be divided between the time zones that development does not stop.

53. Visits to the sites should be promoted.

54. Enhance active involvement in all aspects of development for better understanding with customer.

55. Confidence should be given to both formal and informal meetings among all stakeholders.

56. Data should be shared with the customer at all stages.

57. Software configuration management should be used to manage the different components of the software system.

58. Increased dependency on the partners should be encouraged.

59. Regular meetings with the customer will give better reviews and feedback.

60. Communication among team members should be ensured.

61. Good management among one another, the project team and top management along with the customers.

62. The role of the customer should be expanded to process of development of product along with discussing the important features, prioritizing requirements and user demands from time to time.

63. Closely engage the customers into the development phase.

64. Attempt to understand and improve the current climatic and economic problem of software organizations.

65. Current practices only focus on sustainability for software developers.

66. Practices in software processes to efficiently ensure services, lower costs, and better systems control.

67. The deployment and implementation of software systems for better contribution to sustainable processes.

68. The adoption of agile methodology for sustained software organizations varies with the network physical infrastructure.

69. How to attain present goals without compromising the future generations and for them to achieve their goals.

70. Knowledge management tools should be used to carry out data de-duplication.

71. Install disaster recovery software application.

72. Data de-duplication rightsizing software equipment, storage tiring.

73. The reuse of software modules for knowledge using coding.

74. Manage efficiently control life cycle to avoid data redundancy.

75. Efficient coding applications for software development.

76. Delete old and unused servers from the database.

77. New software systems should operate in energy efficient methods.

78. Understand the effect of software usage on the Environment.

79. Sharing information continuously will increase the agile practices.

80. Reduced delivery schedules and increased return on investments.

81. Individual competency should be streamed with the team requirements for it to perform better and efficiently.

82. Arrange Trainings and meeting for the team to get work together effectively and increase competency levels.

83. Training for the team we can get better results faster.

84. Remote communication with their team to work effectively and efficiently.

85. Frequent visits between team members and customers to maintain collaboration.

86. Focus on risks related to customers, communication, and stakeholders.

87. Deliver final software to the end users quicker than traditional approaches.

88. Involve the customer in agile projects.

89. Extensive work has been done in Risk Management and present work that is needed for further consideration.

90. Customer commitments and presence have full authority over other issues.

91. Study the current approaches and find out new ways of new areas to explore.

92. If risk is planned carefully initially the percentage of failure is reduced.

93. If risk is managed carefully the chances of failure are reduced.

94. Risks associated with high company effect are identified by the departments.

95. Assessed and mitigated the risk properly and communicate to your entire organization.

96. After you have assumed the initial risk assessment then you should put the relevant risk controls for mitigation and monitoring.

97. Sharing of knowledge, expectations and concerns in work ethics should be done through video conferences, emails, and calls.

98. Use latest technology and processes for knowledge sharing and management.

99. Variance analysis should be used when and where it is needed.

100. Establish how new procedures can be introduced into the workforce for smooth information sharing among new and old team employees.

101. Convert tacit knowledge to explicit knowledge by documentation and process description.

102. Implement new domain and technical trainings to update the database and better profiling of the employees.

103. Team level collaboration and knowledge should be shared for adoptability for both large- and small-scale teams.

104. Improved capability of the various vendors of better implementation of SPI certifications, i.e., CMM and CMMI.

105. Improving product quality through proper monitoring.

106. Better competition through service provision.

107. Better ways of providing interaction among team players for better tacit knowledge sharing.

108. Hiring process should be based on good job skills.

109. Provision of best services design and execution.

110. Follow strict time development schedule.

111. Establish mutual trust.

112. Offer quality management trainings.

113. Standard RE models should be used for conducting Requirements Engineering phase.

114. Improve client-vendor communication.

115. Ensure that the client requirements of response time, flexibility, usability, and reliability are met.

116. Ensuring that there is provision of global talent, and the delivery models should be world class.

117. Efficient problem solving, better communication and strong collaboration among team members.

118. In different situations the capacity should work effectively, as change management is known to be unpredictable.

119. Team members should have clear set priorities and they should be kept moving ahead to reach other goals.

120. Change in the organization needs to be done in a collaborative way with the old problems being shared with the employees along with the new benefits of the new system.

121. Sudden changes to the working environment should not be introduced as it is rejected, and employees are not able to adjust to it.

122. Release of information change should be done quickly and then the incremental steps involved should be performed.

123. Frequently meet with your change team employees and encouraging them to share their feedback and provide what works and what is not working.

124. Awareness to the employees that this is a learning curve, and it will have its share of questions, concerns and suggestions which are all entertained.

125. Fostering trustful ties with the client.

126. Additional favors in development of mutually beneficial partnership.

127. Apply effort and other resources for maintaining on going relationships.

128. "Future orientation" plans both long range and short range should be discussed.

129. The other factors related to customer satisfaction and service along with both financial performance, internal business and growth should be considered.

130. Better learning requires to understand and listen to other ideas and opinions.

131. Confidence in the working principle of Agile would be so suitable and gaining acceptance of its approach is increased.

132. Various learning experiments vided insights into the problems and who they were mitigated.

133. To have better communication and contact constant contact was maintained in a working environment.

134. By using and creating project through Agile more confidence was given to the development team members.

135. Pilot projects success and failures tend to give stakeholders information at an early stage about the project.

136. If the pilot project was not received well but overall, the idea is good and should be pushed, gives the project team staff an opportunity to adopt a new strategy for better success.

137. Agile software development has specific management roles which keep a hold on both the unique and difficult processes.

138. Following roles and techniques for better planning and flexibility and learning.

139. Have a better organization and learning mechanism to deal with the complexity and unpredictability of agile software projects.

140. To better understand the specific management functions a unique and complex process and its activities should be implemented.

141. Training and learning approached should be increased in terms of skills diversity.

142. National Training Framework should be adopted which is linked to accredited training.

143. Requirements management is all about learning and documenting the work to be performed by the project and ensuring compatibility with resources.

144. Resolving, analyze, specify, validate, and manage software requirements.

145. They turn the business potential into actual competitiveness on the market.

146. They define the requirements in terms of goals which are well understood by the stakeholders.

147. Deals with crosscutting requirements systematically and this can be integrated by agile software development methodologies.

## Appendix C

| | **Science Direct** |
|---|---|
| P1 | A survey study of critical success factors in agile software projects in former Yugoslavia IT companies |
| P2 | Identifying Some Important Success Factors In Adopting Agile Software Development Practices |
| P3 | The lean gap: A review of lean approaches to large-scale software systems development |
| P4 | The impact of inadequate customer collaboration on self-organizing Agile teams |
| P5 | A comparison of issues and advantages in agile and incremental development between state of the art and an industrial case |
| P6 | A comparison of issues and advantages in agile and incremental development between state of the art and an industrial case |
| P7 | Challenges of shared decision-making: A multiple case study of agile software development |
| | **IEEE** |
| P8 | A survey study of critical success factors in agile software projects |

| | |
|---|---|
| P9 | It's not the pants, it's the people in the pants" Learning's from The Gap Agile Transformation – What Worked, How We Did it, and What Still Puzzles Us |
| P10 | Agility in a Large-Scale System Engineering Project: A Case-Study of an Advanced Communication System Project |
| P11 | The Virtual Agile Enterprise: Making the Most of a Software Engineering Course |
| P12 | Distributed Agile Development: Using Scrum in a Large Project |
| P13 | Governance of an Agile Software Project |
| P14 | Construction of an Agile Software Product-Enhancement Process by Using an Agile Software Solution Framework (ASSF) and Situational Method Engineering |
| P15 | Scaling Agile: Finding your Agile Tribe |
| P16 | Experiences on Agile seating, facilities and solutions Multisite environment |
| P17 | Agile Transformation: What is it about? |
| P18 | Inter-team Coordination in Large-Scale Globally distributed Scrum: Do Scrum-of-Scrums Really Work? |
| P19 | Hidden Facilitators of Agile Transition: Agile Coaches and Agile Champions |
| P20 | Applying Agile Methodologies in Industry Projects: Benefits and Challenges |
| P21 | Exploratory Study of Architectural Practices and Challenges in Using Agile Software Development Approaches |
| P22 | Agile Project Leadership – My Top 10 Value Driven Principles |
| P23 | Migrating From SharePoint to a Better Scrum Tool |
| P24 | Enterprise Scrum: Scaling Scrum to the Executive Level |
| P25 | How BMC is Scaling Agile Development |
| P26 | Fast & Predictable – A Lightweight Release Framework Promotes Agility through Rhythm and Flow |
| **Google Scholar** | |
| P27 | A Quantitative Study on Critical Success Factors in Agile Software Development Projects; Case Study IT Company |
| P28 | Adopting Agile Software Development Practices: Success Factors, Changes Required, and Challenges |
| P29 | Investigating Adoption Of Furthermore, Success Factors For Agile Software Development In Malaysia |
| P30 | Success Factors for Building and Managing High Performance Agile Software Development Teams |
| P31 | An Empirical Study into Social Success Factors for Agile Software Development |
| P32 | Success Factors of Agile Software Development |
| P33 | Critical Success Factors in Distributed Agile for Outsourced Product Development |
| P34 | Agile Adoption in IT Organizations |
| P35 | Agile Software Development in Distributed Environments |
| P36 | A Bayesian Based Method for Agile Software Development Release Planning and Project Health Monitoring |
| P37 | A contingency fit model of critical success factors for software development projects :A comparison of agile and traditional plan-based methodologies |
| P38 | Agile in global software engineering: an exploratory experience |
| P39 | Agile Success Factors A qualitative study about what makes agile projects successful |
| P40 | Identify and Classify Critical Success Factor of Agile Software Development Methodology Using Mind Map |
| P41 | Agile principles and achievement of success in software development: A quantitative study in Brazilian organizations |

| | |
|---|---|
| P42 | How Agile Are You Thinking? – An Exploratory Case Study |
| P43 | Agile Software Development Framework in a Small Project Environment |
| P44 | Risks of Agile Software Development: Learning from Adopters |
| P45 | Agile transition and adoption human-related challenges and issues: A Grounded Theory approach |
| P46 | Empirical Investigation on Success Factors in Adapting Agile Methodology in Software Development at Public Organizations |
| **SpringerLink** | |
| P47 | Technical Dependency Challenges in Large-Scale Agile Software Development |
| P48 | Tailoring Agile in the Large Experience and Reflections |
| P49 | Towards Principles of Large-Scale Agile Development |
| P50 | Scaling up the Planning Game: Collaboration Challenges in Large-Scale Agile Product Development |
| P51 | Towards Agile and Beyond: An Empirical Account on the Challenges Involved When Advancing Software Development Practices |
| P52 | Agile Software Development in Practice |
| **ACM** | |
| P53 | Best Managerial Practices in Agile Development |
| P54 | Supported Approach for Agile Methods Adaptation: An Adoption Study |
| P55 | Communication Factors for Speed and Reuse in Large Scale Agile Software Development |
| P56 | Motivations and Measurements in an Agile Case Study |
| P57 | A Case Study on Benefits and Side-Effects of Agile Practices in Large-Scale Requirements Engineering |
| P58 | Inter-team Coordination in Large-Scale Globally Distributed Scrum: Do Scrum-of-Scrums Really Work? |

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
