# Peer review of "Practices of Motivators in Adopting Agile Software Development at Large Scale Development Team from Management Perspective"

_electronics, doi:10.3390/electronics10192341_

Round 1

Reviewer 1 Report

The article presents a study that reviews the literature related to practices and success factors in the context of large-scale software development.

My primary concern is related to the link between the success factors, motivators, and practices. I couldn’t understand how the authors managed to link the practices to the success factors, and I couldn’t understand what the authors understood by motivators since it is never explained. Is it the same as success factor? I think they are entirely different concepts.

Another critical point is the use of arbitrary thresholds to determine important or critical factors and practices. For instance, critical practices are those whose frequency is greater or equal to 30 and different cutting points are used (e.g., %>15, %20, etc.). These thresholds values should be either reasonably justified or removed.

The discussion is missing at the moment. In the discussion section, authors usually interpret the results and give insights on the findings. The authors should answer the research question too. This interpretation is missing at the moment. As it is, the paper shows a systematic mapping study rather than a systematic literature review.

The limitations of the study should be further elaborated by describing the threats to validity and how they were mitigated.

Finally, I think the SLR does not provide enough information about the implementation of the practices. This point requires analysing the contextual factors such as company size, domain, etc., to understand if there are clear patterns. The article does not cover this point.

Detailed comments:

Agile software package program improvement? --> do you mean “agile software development” page 2, line 28

“to extend software package quick” --> to develop or extend software Line 29

Software program improvement --> does it refer to software process improvement or software maintenance.

Large agile methodology --> there is not only one methodology, so this should be revised (line 46)

The need to follow the Software Development Life Cycle (SDLC) used should be avoided (line 70) --> this term SDLC is used here for first time and never explained. If it refers to a particular methodology, it should be explained; otherwise, the sentence is wrong because agile methods also propose a software development life cycle.

Introduction:

In the introduction, the authors talk about success factors and then about Agile methods. I do not see a connection between the first two paragraphs. The authors’ idea is unclear.

The RQ makes a distinction between practices and solutions. Why? This should be clarified.

The RQ also mentions “motivators”, but this concept was never introduced.

The intro is not easy to follow. The concepts are mixed-up, many acronyms are presented and never used afterwards, there is no clear motivation of the research question.

Line 81. “Agile started from 2007 which was called as the version one,” This sentence is incorrect. The agile manifesto dates from 2000.

Line 112. Abrar et al. --> reference is missing.

The authors use the work of Abrar et al. since it describes categories and practices carried out for large scale ASDM adoption from management perspective. What alternative studies have been considered? The list of categories and practices should be presented.

Line 114. By applying the methodology of SLR to our study it will define the missing practices in the Agile transformation. --> Which agile transformation? Do you mean, the less frequent practices that are used for agile transformations?

Research methodology.

Section 3.1. The search string is missing.

Line 130. “the title of the paper and our research title should match.” --> I doubt there are papers that match exactly “the title of the paper”, I think this sentence is wrong.

The inclusion and exclusion criteria should be presented in a Table for better reading.

“Thus, after applying the quality assessment criterion only 82 papers made the final list.” --> How was it done?

Figure 1 is very generic. There are typos such as “Documantation”. In addition, the steps presented in the figure are not described in the text.

Line 135. “We also conducted the inter-rater reliability test to remove the biasness but we found not variance.” --> How was it done?

Line 137. The information about the retrieved papers should be presented in the results section.

The role of the authors should be clarified in each step.

Figure 2 starts from step 4. Why were the previous steps omitted? Clarify or update the figure.

Figure 3 is misleading since the number indicate the categories rather than the total number of studies. The type of chart used is not the most informative one. I’d prefer to read a table with all the frequencies that the authors mention in line 146. This is missing at the moment.

Figure 4. The type of chart used is incorrect. The x-axis shows categories; thus, a categorical chart must be used (e.g., a bar chart)

Line 169. “Those practices will be considered as critical whose frequency was >= 30.” Why did the authors choose this number? It seems arbitrary, and it is not representative of critical practices. I recommend removing this arbitrary classification if it cannot be justified.

Table 1 should be ordered by frequency. The caption of the table is “motivators” whereas the main column is about “Success factors”. These concepts are not the same.

Table 1 should have a new column with the articles mentioning the success factor.

Line 338. “We have identified 146 practices in total” --> Where are these practices listed? The authors should attach supplementary material or an appendix listing all the practices.

The discussion section shouldn’t show the results of the practices. The practices are part of the results too.

The manuscript should be revised entirely regarding the flow of ideas, clarity, and use of English (there are many grammar mistakes and typos).

Author Response

Response to Referee Report

Manuscript ID: electronics-1303068, “Practices of Motivators In Adopting Agile Software Development At Large Scale Development Team from Management Perspective"

Rashid Khan, Muhammad Faisal Abrar, Samad Baseer, Muhammad Faran Majeed, Muhammad Usman, Shams Ur Rahman, You-Ze Cho

August 8, 2021

We would like to thank the Journal name for considering our manuscript and our referees for their careful reviews. We received the reviews and decisions that we should provide a major revision. We would hereby like to resubmit our manuscript with the editorial and reviewer comments addressed. We respond to the specific comments below. In the manuscript, new text is colored BLUE for differentiation.

Reviewer #1

Comments and Suggestions for Authors: The article presents a study that reviews the literature related to practices and success factors in the context of large-scale software development.

We thank the reviewer for the thorough review of our manuscript.

  1. My primary concern is related to the link between the success factors, motivators, and practices. I couldn’t understand how the authors managed to link the practices to the success factors, and I couldn’t understand what the authors understood by motivators since it is never explained. Is it the same as success factor? I think they are entirely different concepts.

We thank the reviewer for highlighting this point. While the success factors can be defined as “the factors, leading towards the successful adoption of ASDM to large scale.” On the other hand, the word “motivators” is interchangeability used as success factors. And practices are the solutions to the success factors and motivators. For the ease of the users, we have corrected and defined these terms in the manuscript where required.

  1. Another critical point is the use of arbitrary thresholds to determine important or critical factors and practices. For instance, critical practices are those whose frequency is greater or equal to 30 and different cutting points are used (e.g., %>15, %20, etc.). These thresholds values should be either reasonably justified or removed.

We thank the reviewer for the suggestion however, the use of these threshold set up a filter to take only those practices that come under the consideration. Removing these thresholds will drastically increase the number of irrelated practices.

iii. The discussion is missing at the moment. In the discussion section, authors usually interpret the results and give insights on the findings. The authors should answer the research question too. This interpretation is missing at the moment. As it is, the paper shows a systematic mapping study rather than a systematic literature review.

We thank the reviewer for this comment. As per reviewer 2, we have combined the discussion and results sections which has significantly improved the discussion regarding the findings. Also, we have added some discussion to further elaborate the research question.

To make manuscript more a systematic literature review we have re-arranged and rewritten some of the paragraphs.

  1. The limitations of the study should be further elaborated by describing the threats to validity and how they were mitigated.

We elaborated validity threats and also provided their mitigations statements. The text is as follow.

In this section, the threats of validity concerning the SLR study have been discussed. By using SLR procedure,we mined practices / solutions for each of the SFs in the adoption of ASDM at large scale from management perspectives, but how valid are our findings? Related to internal validity ever first threat to be, for any study, they have not explicitly mentioned the cause to report solutions of the ASDM transformation factors. We are unable to control this threat. However, we have tried our level best for the purpose to validates the results through interrelated test i.e., checking the results by all authors and decided whether to include the practices or exclude. This detail checking was time consuming, but it gives us the satisfaction to find the result guanine. Furthermore, one possible threat to internal validity is that for any practice/solution in an article, the author may not in fact have described the underlying cause of practices for SFs implementation in ASDM transformation from conventional software developmental methods for any specific response.

 Concerning to the threat of external validity, our sample size is composed of the articles reporting data from diverse countries. We have a full confidence in our results because, we found similarities greater than differences in our outcomes. This provides evidence for generalization. We have conducted our SLR in teamwork and consulted the software engineering research group at University of Engineering & Technology, Mardan (SERG_UETM) for validation of the search strings. with the increasing number of papers in ASDM, our SLR process may have missed out some relevant papers. However, like other researchers of SLR this is not a systematic omission [8]. To deal with subjectivity and researcher biases, we have also done interrater reliability check in every step of the SLR conduction. We do not claim that we have included all digital libraries, so executing our SLR process; it is possible to miss some relevant paper(s). The first reason is abundant papers on ASDM. The second reason is inaccessibility of every digital library due to lack of resources. However, the included digital libraries are more than enough for the synthesis of results in our study. According to other academic’s investigator like [8] and [67] using SLR as a method for data collection, this is not a methodical omission.

  1. Finally, I think the SLR does not provide enough information about the implementation of the practices. This point requires analysing the contextual factors such as company size, domain, etc., to understand if there are clear patterns. The article does not cover this point.

We thank the reviewer for this comment. As it can be perceived from Figures 3 and 4 that the domain of our research is continent wise. Selecting company size etc. as a contextual factor brings the use of micro analysis. Which gets out of the scope of our study. However, this is a very good point for conducting future research.

Detailed comments:

  1. Agile software package program improvement? --> do you mean “agile software development” page 2, line 28

Corrected

  1. “to extend software package quick” --> to develop or extend software Line 29

Corrected.

iii. Software program improvement --> does it refer to software process improvement or software maintenance.

Yes, Software process improvement

  1. Large agile methodology --> there is not only one methodology, so this should be revised (line 46)

Thank you for the comment. The text is corrected.

  1. The need to follow the Software Development Life Cycle (SDLC) used should be avoided (line 70) --> this term SDLC is used here for first time and never explained. If it refers to a particular methodology, it should be explained; otherwise, the sentence is wrong because agile methods also propose a software development life cycle.

Sentence Removed

Introduction:

  1. In the introduction, the authors talk about success factors and then about Agile methods. I do not see a connection between the first two paragraphs. The authors’ idea is unclear.

Thank you for this comment. The introduction section is revised and re organized for readability purpose.

  1. The RQ makes a distinction between practices and solutions. Why? This should be clarified.

Thank you, we have clarified this comment in comment-i.

iii. The RQ also mentions “motivators”, but this concept was never introduced.

We have clarified this comment in comment-i.

  1. The intro is not easy to follow. The concepts are mixed-up, many acronyms are presented and never used afterwards, there is no clear motivation of the research question.

We have revised the introduction section as per comment-1 of the introduction.

  1. Line 81. “Agile started from 2007 which was called as the version one,” This sentence is incorrect. The agile manifesto dates from 2000.

Corrected.

  1. Line 112. Abrar et al. --> reference is missing.

corrected

vii. The authors use the work of Abrar et al. since it describes categories and practices carried out for large scale ASDM adoption from management perspective. What alternative studies have been considered? The list of categories and practices should be presented.

We thank the reviewer for highlighting this point. we have included the list of categories as Table 1.

viii. Line 114. By applying the methodology of SLR to our study it will define the missing practices in the Agile transformation. --> Which agile transformation? Do you mean, the less frequent practices that are used for agile transformations?

We thank the reviewer for this comment. As our study is SLR based, we include all the practices that come under the umbrella of SLR approach. For better understanding we have included the following text after the above sentence.

As the SLR covers all the possible publications that include conference, preceding, international journal papers , reports case studies , thesis and dissertation etc.

Research methodology:

  1. Section 3.1. The search string is missing.

We have included the search string as follows 3.1.4

("Agile Method" OR "Agile Software Development" OR "Agile Method OR Agile Development") AND ("Large Development Team" OR "Large Development Team" OR "Large Development Team") AND ("Incentives" OR "motivators" OR "Factors" OR "Success The factor" OR "positively affects" OR “promoters" OR  "supporters" OR "key factors") AND ("Practices" OR "Solutions).

  1. Line 130. “The title of the paper and our research title should match.” --> I doubt there are papers that match exactly “the title of the paper”, I think this sentence is wrong.

The statement is revised as

the keywords in the title of the paper and the keywords in our research title should match.

iii. The inclusion and exclusion criteria should be presented in a Table for better reading.

We have created the table as table 1.

  1. “Thus, after applying the quality assessment criterion only 82 papers made the final list.” --> How was it done?

The final list included 82 papers because the data extracted only from these 82 sources.

  1. Figure 1 is very generic. There are typos such as “Documantation”. In addition, the steps presented in the figure are not described in the text.

We have corrected the typos. Also, the generic nature of the figure provides overall insight of the steps carried out during the SLR process. The steps included in the figure are thoroughly described in Section 3.

  1. Line 135. “We also conducted the inter-rater reliability test to remove the biasness but we found not variance.” --> How was it done?

We thank the reviewer for this comment. The process followed for the above sentence is thoroughly described in Section 3.3 in which two phases are defined for the selection of primary sources.

vii. Line 137. The information about the retrieved papers should be presented in the results section.

We have added the list as Appendix A.

viii. The role of the authors should be clarified in each step.

As per our understanding of this comment. This comment is highlighting the roles of the authors of the very manuscript describing the role of each author at every step would bring a sort of difficulty because it is a collective effort by all the authors.

  1. Figure 2 starts from step 4. Why were the previous steps omitted? Clarify or update the figure.

We thank the reviewer for highlighting this point. The following text is added to Section 3 for the purpose to remove this ambiguity.

The following steps were derived from the guidelines for performing SLRs in software engineering [2] and applied as a procedure in systematically searching and selecting the relevant studies.

  1. Define the research objective.
  2. Conduct several example searches; review the scopes.
  3. Define the search string; identify inclusion and exclusion criteria.
  4. Conduct an initial search.
  5. Review the title, abstract, and keywords of the initially retrieved studies.
  6. Revise inclusion and exclusion criteria; select potentially relevant studies.
  7. Remove duplicate studies.
  8. Review potentially relevant studies selected; discuss any issues.
  9. Review the entire content of initially selected studies (including the references section for identifying the studies that are potentially missed); identify relevant ones.
  10. Review relevant studies selected; discuss any issues.
  11. Identify the final set of relevant studies.
  12. Figure 3 is misleading since the number indicate the categories rather than the total number of studies. The type of chart used is not the most informative one. I’d prefer to read a table with all the frequencies that the authors mention in line 146. This is missing at the moment.

As per the comment of the reviewer, we have converted the figure to a graph to easily depict the results.

  1. Figure 4. The type of chart used is incorrect. The x-axis shows categories; thus, a categorical chart must be used (e.g., a bar chart)

As per the suggestion of the reviewer, we have converted the categorical chart into a bar chart. That looks more understandable.

xii. Line 169. “Those practices will be considered as critical whose frequency was >= 30.” Why did the authors choose this number? It seems arbitrary, and it is not representative of critical practices. I recommend removing this arbitrary classification if it cannot be justified.

As every research has its own criteria for the selection of the frequency. The selection of this frequency is due to the fact that greater the number of frequency greater will be the inclusion of irrelevant practices as vice versa.

xiii. Table 1 should be ordered by frequency. The caption of the table is “motivators” whereas the main column is about “Success factors”. These concepts are not the same.

As per the suggestion of the reviewer we have normalized the term and ordered the terms by frequency.

xiv. Table 1 should have a new column with the articles mentioning the success factor.

We have included the new column.

  1. Line 338. “We have identified 146 practices in total” --> Where are these practices listed? The authors should attach supplementary material or an appendix listing all the practices.

We have mentioned these practices in the tables. The list is also enlisted as Appendix B.

xvi. The discussion section shouldn’t show the results of the practices. The practices are part of the results too.

As per Reviewer 1 Comment-3, we have revised the discussion section.

xvii. The manuscript should be revised entirely regarding the flow of ideas, clarity, and use of English (there are many grammar mistakes and typos).

We thank the reviewer for the suggestion. We have revised our manuscript by native English speaker, and it has significantly improved the presentation of the manuscript.

Reviewer 2 Report

Pleade find all my comments in the anexed file.

Author Response

Response to Referee Report

Manuscript ID: electronics-1303068, “Practices of Motivators In Adopting Agile Software Development At Large Scale Development Team from Management Perspective"

Rashid Khan, Muhammad Faisal Abrar, Samad Baseer, Muhammad Faran Majeed, Muhammad Usman, Shams Ur Rahman, You-Ze Cho

August 8, 2021

We would like to thank the Journal name for considering our manuscript and our referees for their careful reviews. We received the reviews and decisions that we should provide a major revision. We would hereby like to resubmit our manuscript with the editorial and reviewer comments addressed. We respond to the specific comments below. In the manuscript, new text is colored BLUE for differentiation.

We thank the reviewer for the thorough review of our manuscript.

  1. INTRODUCTION:

This section should be revised because it is very difficult to read and understand it. Authors start taking about “Success Factors” and then about “agile methods” but these two concepts are not clearly related.

We thank the reviewer for this comment. We have revised the Introduction section, as per comment-1 of the Reviewer-1  of the introduction.

Some comments:

  1. The first sentence of the introduction needs to be rewritten. It is not clearly presented and just according to the Wikipedia it seems that is the other way round: “The concept of "success factors" was developed by D. Ronald Daniel of McKinsey & Company in 1961. The process was refined into critical success factors by John F. Rockart between 1979 and 1981.”. In the text, authors should refer to the complete name of the referenced author as Wikipedia does.

We thank the reviewer for highlighting this point. We have revised the sentence and provided proper reference from authentic source. The text is as follows.

The concept of "success factors" was developed by D. Ronald Daniel of McKinsey & Company in 1961 [5]. The process was refined into critical success factors by John F. Rockart between 1979 [6] and 1981 [7]. The idea of Success Factor (SF) or motivators was first introduce by Rockart in which he uses a specific instrument for statistical data identification which are the basic scope of top management [8]. Daniel proposed the concept of success factors in the management literature based on the Rockart concept [6].

  1. In line 18: the sentence “Good characteristics with environmental uncertainty are very important for customer satisfaction” which characteristics are the authors referring to? of what?

We thank the reviewer for this comment. We have revised the sentence as,

Good characteristics such as “traditional organizational culture” , “ Team Collaboration” and “Project management”  with  environmental uncertainty are very important for customer satisfaction

iii. Lines 22-23 need to be rewritten too. Authors use the adverb “however” but the second sentence doesn´t introduce an idea which is the opposite of the one presented in the first sentence.

We have re-written the sentence and made it more understandable.

  1. line 26: “The words Agile methods have been around for over 10 years, ...” The concept of agility has been around for much more than 10 years. The agile Manifesto was published in 2001.

We have re-written the text as “for two decades”.

  1. Line 28: “what agile software package program improvement is”. I do not understand what the authors wanted to say. This sentence needs to be rewritten.

Corrected.

  1. Line 29: “purpose” is a name, not a verb, so, I think that the sentence lack of a suitable verb.

Corrected as follow.

In truth, there is still no entire settlement on what agile software development methods is, but actually agile Techniques’ purpose is to answer a requirement to extend software package quick, in surroundings of quickly changing requirements.

vii. Line 31-33: “Also, […] customer end user.” This sentence does not make sense.

We have removed the sentence as it was ambiguous and unnecessary.

viii. Line 34. “To coping with” or “to cope with”?

Corrected.

  1. Line 37. The preposition that goes with the verb “allow” is “to”. Allow someone to do something.

Corrected.

  1. Line 46. Authors are talking about XP but the reference used [9] is not about XP.

Corrected.

  1. Line 50. incantation? The sentence does not make sense.

Revised and corrected.

  1. LITERATURE REVIEW:
  2. Line 81. Authors state that “Agile started from 2007” but the Agile Manifesto was published in 2001.

 Corrected.

  1. Line 84. The sentence “In a recent survey done in both the works” Is not correct.

Corrected as

In a recent survey based on both the research shows that out of four thousand participants only 62% stated that they have more than 100 developers in their software house.

  1. Line 86. The remaining of 62% (introduced in the previous sentence) is 38%. But both data are not related since they reveal difference information. The first figure is about the number of developers and the second one has to do with the use of agile practice. Consequently, “the remaining” can not be used.

Corrected

iii. Line 99-100. The goal of the paper is not clearly stated. Authors say:” we aim to fill out the gap between the SLR for LSAD team and projects from management angle.” Is there a gap between a Software Literature Review and projects? It does not make sense.

Corrected as

in this paper our aim is to fill out the gap of projects and teams for LSAD from the management perspectives through SLR.

  1. Line 101. “Our stud”, do you want to say “study”?

Corrected.

  1. RESEARCH METHODOLOGY:
  2. I miss the classical reference to B. Kitchenham who published the guides to deal with a SLR.

We have edited the classical reference to B. Kichenham at appropriate place.

  1. Line 109. It is ambiguous to say “Abrar et al.”. Authors should clearly state who of them have participated in each part of the study.

As per our understanding to the comments is highlighting the roles of the authors of the very manuscript describing the role of each author at every step would bring a sort of difficulty because it is a collective effort by all the authors.

iii. Line 110. Research questions? In plural? Only one question has been defined!

Corrected.

  1. Line 114. “By applying the methodology of SLR to our study it will define the missing practices in 2 the Agile transformation.“ I understood that the goal of the SLR was to identify the most suitable agile practices for each success factor, is that right?

Yes, we find most suitable agile practices for each success factor.

  1. Line 121. SLR techniques? Do you refer to “agile” techniques?

Corrected and Revised.

  1. Line 122. The sentence is uncomplete. It can not finish with “:”

Corrected and Revised.

vii. Line 125. The search string is not explicitly shown.

We have included the search string at Section 3.1.4.

viii. Line 129. “The criteria” I suppose that it refers to “inclusion criteria”, but in the previous sentence authors are talking about inclusion and exclusion criteria.

Yes, we are talking about the inclusion and exclusion criteria.

  1. Line 141-142. The process for quality assessment has not been explained. The quality criteria have not been stated and the ranking of the selected papers according to the quality criteria is missing. I do not understand the relationship between the study [21] and the quality assessment.

We have provided the research on the basis of which quality criteria was selected. This quality criteria participated in selection of papers for our research.

  1. Line 144. The sentence “The data extraction was done by a team of researchers’ who were responsible for the data extraction.” expresses an obviousness. Again… which of the authors have participated in this part?

Corrected and revised.

  1. Line 147. “Critical factors methodology”? Methodology for identifying critical factors? But… Are you looking for agile practices in this SLR?

Yes, Corrected and Revised.

xii. Line 148. Types? Type of what? Do you refer to “types of practices”?

Corrected.

xiii. Line 150. The title of the section is duplicated at the beginning of the first sentence.

Corrected.

xiv. Line 151. It says that the occurrence of each agile practice has been counted as it shown in Figure 3. But this figure shows a different information.

Corrected and provided appropriate data in the appendix.

  1. Line 154. It refers to Figure 1 to show the classification of the final selected papers but Figure 1 shows the SLR process.

Corrected and provided appropriate data in the appendix-a.

xvi. Line 160. The comment “The figure depicts…” is out of place because the figure that it refers to is Figure 4 and not Figure 3.

Corrected.

xvii. Line 164. Instead of Figure 2 is Figure 3.

Corrected.

  1. RESULTS:
  2. Line 171-172. It is said that “This section demonstrates the outcomes of the SLR i.e., the practices/solutions for implementing Motivators for scaling ASDM” but in fact only the motivators are presented in this section. The practices associated to each success factor are not presented until the Discussion section. Authors should clarify that in this section the motivators or success factors are going to be summarized because they come from another SLR [1]. The reference to the previous work only appears in the tittle of Table 1. This reference should appear in the text.

To provide a differentiation and better understanding to the reader we have separated the outcomes of the SLR from the practices associated to each success factor in different sections.

  1. Line 213. & refers to “and”?

Corrected

iii. Line 232-233. I do not understand the meaning of this sentence: “the quantity of defects toughened by the client will be Associate in Nursing external live of the wares”

Revised and corrected. 

  1. Line 247-248. The sentence “Existing techniques particularly attention on realizing sustainability when software program experts improve software.” This sentence needs to be revised. Attention is a noun not a verb. The whole 4.10 section needs to be revised.

Corrected.

  1. Line 251. Why “Influences” starts with uppercase?

Corrected.

  1. Line 263. sufficient enough?

Corrected.

vii. Line 276. “Risk management Distributed Agile Development (DAD)” needs a verb.

Corrected.

viii. Line 282-286. One sentence takes 5 lines!! It is difficult to understand.

Corrected

  1. Line 301. “Further” with uppercase? This sentence needs a verb.

Corrected

  1. Line 312. The acronym SDP has not been defined. Although it is obvious the meaning, all the acronyms must be defined the first time that the word is written.

Corrected.

Results:

  1. Section “4 Results” could be joined with the next section since it does not show the result from the current SLR but the results of a previous research. The current paper should clearly state the agile practices associated with each “success factor” since this is the main contribution.

We have combined the section 4 and section 5 as Result and Discussion.

  1. DISCUSSION
  2. Line 343. %age?

Corrected.

  1. Line 354. You have to remove the dot since this sentence is not part of the previous list.

Corrected

iii. Line 374. Illustrious? Illustrates?

Corrected

  1. Line 383. It says “four” instead of “five”

Corrected

  1. Line 384. This line should be removed.

Removed and Revised.

  1. Line 393. Memorable is not a verb.

Corrected

vii. Line 395. Again, the first item of the list should be removed. Please, revise the coherence between the text, the information in the tables and the lists. I am not going to mark any more mistakes in this vein. I am not sure it is necessary to duplicate the information between the text and the tables since the text does not introduce more information than the given in the tables. The section could be organized around the motivators or success factors, one subsection for each motivator. In each subsection first the motivator is briefly explained and then the practices to achieve this SF are listed and explained.

We have religiously followed this very comment of the reviewer and rearranged the text for better understanding. This has significantly improved the readiest experience of the manuscript.

viii. Figure 5 and Figure 6 show the information in a different format. In the previous “success factors” the practices are shown in a tabular form. I think that all the information should be presented by using the same format for uniformity.

As per the comment we have changed the figures to tables.

  1. CONCLUSIONS:
  2. Line 579 says “After applying the criterion, it was reduced to 99.” In the paper this reduction has not be explained neither the criterion defined. It is surprising that although the practices are obtained from the selected papers in the SLR, authors do not show the correlation between them. It is not showed which practices come from each selected paper.

We thank the reviewer for highlighting this point. The criterion mentioned about has already been presented in the form of inclusion /exclusion criteria and quality of the paper criteria.

Round 2

Reviewer 1 Report

The authors have addressed most of my comments, and the quality of the paper has improved considerably.  

Author Response

Response to Referee Report

Second Revision

Manuscript ID: electronics-1303068, “Practices of Motivators In Adopting Agile Software Development At Large Scale Development Team from Management Perspective"

Rashid Khan, Muhammad Faisal Abrar, Samad Baseer, Muhammad Faran Majeed, Muhammad Usman, Shams Ur Rahman, You-Ze Cho

August 22, 2021

Reviewer #1

Comments and Suggestions for Authors: The authors have addressed most of my comments, and the quality of the paper has improved considerably.

We would like to thank our referees for their careful reviews. 

Reviewer 2 Report

Please, find enclosed my comments in the file that has been attached.

Author Response

Response to Referee Report

Second Revision

Manuscript ID: electronics-1303068, “Practices of Motivators In Adopting Agile Software Development At Large Scale Development Team from Management Perspective"

Rashid Khan, Muhammad Faisal Abrar, Samad Baseer, Muhammad Faran Majeed, Muhammad Usman, Shams Ur Rahman, You-Ze Cho

August 22, 2021

We would like to thank our referees for their careful reviews. We received the second reviews and decisions that we should provide a major revision. We would hereby like to resubmit our manuscript with the editorial and reviewer comments addressed. We respond to the specific comments below. In the manuscript, new text is colored RED for differentiation.

Reviewer #1

Comments and Suggestions for Authors: The authors have addressed most of my comments, and the quality of the paper has improved considerably.

Reviewer #2

I recommended you that the English should be revised but the paper still has many grammatical errors and sentences that are difficult to read. The text should be checked by a native speaker.

We were encouraged by the reviewer to check the manuscript text for grammatical and sentential error, for this purpose we have again thoroughly revised each section of the manuscript. This has significantly improved the readership of the manuscript.

I have seen that authors have improved the organization of the paper by joining Results and Discussion sections. I think that this fact has improved readiness. Nevertheless, I think that the paper should not be published as it is. The introduction still remains unclear, and this section is one of the most important part of the paper since it has to motive people to read.

We thank the reviewer for this comment. Indeed, the Introduction section is the most important part of the paper. In this regard, we have revised the Introduction section, and this has considerably improved this section. We still welcome the comments from the reviewer for any further improvement.

Abstract.

  1. Line 6. I think that the verb is missing. “methodologies applied” or “methodologies are applied”

Corrected

  1. Line 7. Authors say “those factors” as if they were previously presented. They should explicitly introduce the concept of “success factors” and explain that the practices are going to be linked to the success factors identified in a precious work. It is not needed to say that the identification was by means of a SLR because the interested readers can consult the previous research if they want for more information.

We thank the reviewer for the comment. We have revised this portion of the abstract. We have already explained the concept of success factor in the Introduction section.

  1. Introduction

Although authors have revised this section, I think that it remains unclear. Sentences such as the one in lines 34-35 (“This section explained the concept of the word Agile and Agile Software Development Methods at abstract level. The next paragraph is about the Success factors.”) or in Lines 55-57 are not needed since this information can be implicitly stated in the text. This information has to do with the authors´ organization. The problem that I see is that each paragraph introduces a concept: 1) Agile Methods; 2) Success Factors; 3) what projects are considered large-scale projects, but the interest of combining all these concepts is not explained. The justification of the interest and relevance of the paper is not explicitly explained.

We have revised the Introduction section and tried our best to syntactically introduce the concepts in well-ordered paragraphs. This makes the relevance of the concepts more explicit than the previous version.

Lines 20-21. Something is wrong in the sentence “A number of the various agile strategies, eXtreme Programming (XP) improve the practices, of agile success factor in adoption at large scale development team”. If we remove the sentence between comas the remaining sentence makes no sense.

Corrected.

Line 21. Authors have changed the order of the paragraph in the introduction section and they have not realized that they talk about “success factor” in Line 21 but this concept has not been introduced.

We have revised the text.

Lines 21-22. “The uses…” the noun is without “s” “The use…”. The noun is singular so the verb “allow” need an “s”.

Rephrased and corrected.

Line 23 expresses an obviousness.

Rephrased and corrected.

Lines 24-25 introduce the agile software development characteristics but at the end of the sentence the reference [2] is used. This reference has to do with the SLR guidelines not any agile reference.

Line 27. Maximum famous? The most famous?

Rephrased and corrected.

Lines 27-31 introduce XP, so the classical reference to XP has to be used instead of [3]. The classical XP reference is the book written by Kent Beck and Cynthia Andres “Extreme Programming Explained: Embrace Change”

Corrected.

Lines 32-34. “This paper is about the practices of the success factors in the adoption of Agile Software development methodologies at large scale from management perspectives.” The practice of the success factors? I though it was about the selection of the suitable agile practices to achieve the success factors needed for a large-scale projects. This explanation should be given after the paragraph which introduces the term “success factor”.

Replaced and corrected.

Line 37. It seems that authors start a literal definition because they use “ (open but not ended). If so, the reference from which the definition is taken should be indicated. If it is not literally taken from anywhere then the “ is not needed.

Corrected.

Line 50. “if they do not pay attention” to what?

Revised and corrected.

Line 52. “such as” is used when you are going to introduce an example, in your sentence would be the things that managers need to focus on. It is not the case.

Revised and corrected.

  1. Line 62. Is Reference [2] right? [2] is SLR guidelines and this paragraph has to do with largescale projects.

Corrected.

  1. Line 64. A dot is needed at the end of the sentences, before “In the work…” since it is a new sentence.

Corrected.

  • Line 87. “as” could be removed.

Corrected.

  1. Literature Review
  2. This section could be integrated in the introduction since it justifies why a SLR is needed. This justification should be the final paragraph of the introduction. First of all, the concepts of Agile, Success Factors and large-scale projects are introduced and then the reason why the research is needed should be explain.

We thank the reviewer for this comment. We have integrated the LR section in the Introduction section. This improved the readership as well the concepts.

  1. Line 93. “on both of the research”??

Corrected.

  • Line 97. “…for environment for creation”? The reference of the survey this paragraph is talking about is missing.

Corrected.

  1. Line 107. Something is wrong in the sentence “As we move forward the research on ASDM is being developed for creating an SLR” The subject of “is being” is missing. The reference for the SLR should be [2] and not [19-21].

Revised and corrected.

  1. Line 110. If you say “between” is because there are two things implied. You say “between ADSM for large scale development environment” and?

Revised and corrected.

  1. Line 110. Focus from? It should be “focus on”

Corrected.

  1. Research Methodology
  2. Line 115. The reference [2] should be used when you introduce the SLR instead of saying that you are “following the leading scientists in this approach mentioned in [16–18,22].”

Corrected.

  1. Line 118. Instead of “et. al" should be “et al.”

Removed it was irrelevant information as mentioned in comment iii of this section.

  • Lines 118-121. All this information is irrelevant since it comes from a different paper.

Removed it was irrelevant information as mentioned in comment iii of this section.

  1. Line 121. “Then he classifies …” if you refer to Abrar et al. then it is “they” instead of “he”.

Removed it was irrelevant information as mentioned in comment iii of this section.

  1. Lines 123-124. “By applying the methodology of SLR to our study it will define the missing practices in the Agile transformation.” Sure? This result of the SLR is not explored in the paper as a conclusion. It is supposed that you are going to find which agile practice could be used to achieve each success factor, isn´t it? What “missing practices” are you referring to? In any case, you could find a success factors which remain uncover for an agile practice. If so, you should explain better what you want to say.

Rephrased and corrected.

3.1.1 Search String

Perhaps, it is not necessary three different subsections for this issue. It is well-known that the search string must be adapted depending on the database search engine. The complete search string would have been sufficed noting the need of adapting it.

We thank the reviewer for this comment. Following the SLR research guidelines, we have put all the search strings that are implemented during a search process. This creates ease to the readers to go through all the search strings necessary for adopting SLR.

3.2.1 INCLUSION CRITERIA AND EXCLUSION CRITERIA

  1. The title could have been “Inclusion and exclusion criteria”

Revised and corrected.

Table 1 is not referenced from the text.

Corrected.

  1. Line 199. At the end of the sentence is said “The criteria” and it should be “The inclusion criteria”

Corrected.

  • Line 201. “Research title should math”? do you refer to the “search string”? Instead of “should match” do you refer that the candidate paper should include the terms stated in the search string?

We have added the following sentence.

“Our stress is on the matching of the keywords/major terms that appear as a result of the search string. These results will provide a list of publications. The first step is to read the paper’s title and apply the inclusion/exclusion criteria. If the paper’s title matches with major terms of our research question, then the abstract should be read for confirmation of the relevancy by applying the inclusion criteria. Contrary to this, if the search string results do not match even a little with the major terms in our research question, then apply exclusion criteria.”

  1. Line 201. Something is wrong in the sentence: “the papers title match a little then see the abstract and included then if relevant otherwise exclude it.” The conditional form is needed.

Rephrased and revised.

We have added

3.3 Selecting primary sources

  1. Line 204. A dot is needed after “two parts”. “In the initial phase…” starts a new sentence.

Corrected.

  1. - The selection[…] was consisting? Or The selection[…] consisted?

Corrected.

  • - In the initial phase of selection was done? Or In the initial phase, the selection was done

Corrected.

  1. Line 208. “Only 95 papers out of the selected 4069 qualified the inclusion criteria.” This sentence does not match with Figure 2 where the result of Step 7 (apply inclusion and exclusion criteria) is 366 studies. There are some spelling mistakes in the Figure “defferent” instead of “different” and “indusion” instead of “inclusion”.

Corrected.

  1. Line 209. “in” is not needed at the beginning of the sentence. You only have to say “Figure 2 depicts…”

Corrected.

3.4 Publication quality assessment

  1. Line 212. The first sentence “The results gained from the study conducted in [8] were used for publications.” makes no sense since this section has to do with the quality assessment of the new SLR not the previous one. I do not understand why we need the result from [8].

Corrected.

  1. Line 214. “of at”?

Corrected.

  • Line 222. It is not expressed as a question as the rest of the items.

Corrected

  1. Line 223. It is not an item of the list. It should start a new paragraph.

Corrected

  1. The quality assessment is not complete. It is supposed that each paper is scored depending on the answer given to each question. So, the minimal score needed to past the quality assessment must be stated. That way, you can analyse whether all the candidate papers have achieved the minimal score or not.

Corrected.

3.5 Data extraction

  1. Line 226. “till” is a colloquial form, you must use “until”

Corrected

  1. Line 227. “Practice” Why uppercase?

       Corrected

3.6 Data analysis

  1. Authors refer to Appendix 6 that does not exist.

We have cited it correctly but unfortunately, there is an issue with the journal’s latex template.

  1. Lines 232-234. Authors say: “The classification of the final selected papers has been done on different categories based on the continent as a way to find out the development of green and sustainable software.” I do not understand. Now, after seven pages the term “green and sustainable software” appears as the basis for the classification!! Then … the success factors??? The agile practices??? These are the two key ideas of the research question.

Revised and corrected.

  • Line 243. When authors say: “The figure depicts….” Refers to Figure 3 and not to Figure 4 which is the last referenced figure.

Corrected.

4 Results and discussion

  1. Lines 255-258 are written in a repetitive form.

Corrected.

  1. Line 260. Authors say that the term “motivators” is synonym of “critical Success factors”. In the introduction it is said that the synonym is “Success factor”. The term “critical success factor” has not been defined. What is the difference between “success factor” and “critical success factor”. If it is the first time that this term appears you should define the acronym CSF that is used later.

Corrected.

  • Line 272. For success in order to succeed!!

Corrected.             

  1. Table 2 shows the papers selected in [8]. In this paper, authors should show the papers selected in this new SLR. I doubt if authors have conducted a new SLR or they have used the same papers identified in the previous research.

We thank the reviewer for the comment. We have corrected the Corrected against the cited reference. Also, we conducted new SLR that are shown from Table-3 to Table-23.

  1. Line 293 “by follow the following”!!

Corrected.

  1. Line 329 “that most”  “that the most”

Corrected

  • Line 336. “to behavior” Behavior is a noun, not a verb.

Corrected.

  • Line 339. After a dot you should use uppercase.

Corrected

  1. Line 341. Specs? Or specifications?

Corrected

  1. Line 342. Person requirements?

Corrected

  1. Line 377. SLA has not been defined.

Corrected.

  • Line 396. “eminet” is an adjective not a verb.

Corrected.

  • Line 461. It refers to “Chau and Maurer” but the reference is missing.

Corrected.

  • Line 477. “… as the mentioned factors are not sufficient”. Which factors are you referring to?

Removed the irrelevant text.

  1. Line 482. “From the Table 14 it is illustrated”? or Table 14 illustrates?

Corrected

  • Line 490. Seeding visit?

Revised and corrected.

  • Line 500. Something is wrong in this sentence: “Risk management Distributed Agile Development (DAD) and characterize them dependent on the distinguished criteria.”

Revised and corrected.

  • Line 521. Non?

Corrected.

  • Line 524. 13 practices? Table 17 contains 7.

Corrected

  1. Line 533. It? Does it refer to “pair programming”?

Yes, off course.

  • Line 561 “four practices” but five practices are listed in Table 19.

Corrected.

  • Lines 573-574. I do not understand the sentence.

Corrected.

  • Line 583. You say “seven practice” but only four are listed.

Corrected.

  • Line 601. You say “three practice” but five are listed.

Corrected.

  1. Study limitations
  2. Line 652. Guanine?

Corrected.

  1. Line 660. “with” should start with uppercase after a dot.

Corrected.

  • Lines 671-672. “the criterion-the threshold-”?

Corrected.

  1. Line 693. “practices CSFs” makes no sense. Do you refer to practices for achieving the CSFs?

Corrected.

References

  1. [2] The author is Barbara Kitchenham no “Keele, S.; others”. Keele is the name of the university.

Corrected.

  1. [1] and [17] are the same reference.

Corrected.

Round 3

Reviewer 2 Report

I consider that the paper has been notably improved since the last version. Nevertheless, I still have some minor comments:

Abstract – Line 7. Authors talk about “those factors” referring to information that has not been presented. Instead of “those” perhaps they should used “the” and explain which factors are talking about “the factors needed to success in …”

Introduction – Lines 56 – 60. Be careful with the numbers of authors of the references. Sometimes you say “the authors of” when the reference contains just one author, such as [12], [1] and [13]. In line 57 you forgot the “of” before reference [14]. In line 59, reference [9] has more than one author and the “s” is needed.

Line 77. It is the first time that the acronym SLR appears and must be defined.

Smaller Search-Substrings – Line 146-148 the passive form is needed. The “Boolean operators used […] OR operator has been used / was used …”

Inclusion and exclusion criteria – Line 167. The inclusion and exclusion criteria should be different, so, in this sentence you are listing the inclusion criteria not both.

Publication quality assessment. Line 184. I think that the word “only” is not needed since 82 papers is the total number of papers that have passed the inclusion/exclusion criteria. So, the total selected papers have passed the quality assessment. Not only 82 out of 82.

Line 196-197. The sentence “The question which is mandatory in the practices of the success factor.” seams that it is incomplete. Which question are you referring to?

Data Analysis. Line 210. There is an issue with the latex template again because you refer to Appendix 5 that it does not exist.

Result and Discussion. Table 2. I do not understand why the frequency is calculate out of 58 when the total number of selected papers is 82.

Line 242. “That” could be remove -->“In order to know how”

Line 243. After a dot you should start with uppercase.

Reference [18]. The author of the report is not shown.

Author Response

Response to Referee Report

Third  Revision

Manuscript ID: electronics-1303068, “Practices of Motivators In Adopting Agile Software Development At Large Scale Development Team from Management Perspective"

Rashid Khan, Muhammad Faisal Abrar, Samad Baseer, Muhammad Faran Majeed, Muhammad Usman, Shams Ur Rahman, You-Ze Cho

Sept 1st, 2021

We would like to thank our referees for their careful reviews. We received the third reviews and decisions that we should provide a minor revision. We would hereby like to resubmit our manuscript with the editorial and reviewer comments addressed. We respond to the specific comments below. In the manuscript, new text is colored BLUE for differentiation.

Reviewer #1

Comments and Suggestions for Authors: The authors have addressed most of my comments, and the quality of the paper has improved considerably.

Reviewer #2

Comment-1: I consider that the paper has been notably improved since the last version. Nevertheless, I still have some minor comments:

Reply: We thank the reviewer for thorough review of our manuscript.

Comment-2: Abstract – Line 7. Authors talk about “those factors” referring to information that has not been presented. Instead of “those” perhaps they should used “the” and explain which factors are talking about “the factors needed to success in …”

Reply: We have included some the factors in the abstract because including all the factors was not possible. However, it has produced the sense about the factors which are discussed later in the manuscript at Table 2.

Introduction –

Comment-1 Lines 56 – 60. Be careful with the numbers of authors of the references. Sometimes you say “the authors of” when the reference contains just one author, such as [12], [1] and [13].

Reply -: Corrected.

Comment-2:  In line 57 you forgot the “of” before reference [14].

Reply -: Corrected.

Comment-3:  In line 59, reference [9] has more than one author and the “s” is needed.

Reply -: Corrected.

Comment-4: Line 77. It is the first time that the acronym SLR appears and must be defined.

Reply:  Corrected.

Comment-5:  Smaller Search-Substrings – Line 146-148 the passive form is needed. The “Boolean operators used […] OR operator has been used / was used …”

Reply: Corrected.

Comment-6 Inclusion and exclusion criteria – Line 167. The inclusion and exclusion criteria should be different, so, in this sentence you are listing the inclusion criteria not both.

Reply: Corrected.

Comment-7: Publication quality assessment. Line 184. I think that the word “only” is not needed since 82 papers is the total number of papers that have passed the inclusion/exclusion criteria. So, the total selected papers have passed the quality assessment. Not only 82 out of 82.

Reply: Corrected.

Comment-8: Line 196-197. The sentence “The question which is mandatory in the practices of the success factor.” seams that it is incomplete. Which question are you referring to?

Reply: We have revised the following sentence as,

“The question which is mandatory in the data extraction form for finding the practices of the success factor”.

Comment-9: Data Analysis. Line 210. There is an issue with the latex template again because you refer to Appendix 5 that it does not exist.

Reply: We request the Editor to look for this error in the Latex template of the journal.

Comment-1: Result and Discussion. Table 2. I do not understand why the frequency is calculate out of 58 when the total number of selected papers is 82.

Reply: The Table-2 is based on our previous study which included 58 research publications, while our current study includes 82 finally selected papers.

Comment-2: Line 242. “That” could be remove -->“In order to know how”

Reply: Corrected.

Comment-3: Line 243. After a dot you should start with uppercase.

Reply: Corrected.

Comment-4: Reference [18]. The author of the report is not shown.

Reply: We have replaced the reference with the updated report. This solved the issue.

We thank the editor and the reviewers for their deep and thorough attention to our manuscript. The feedback is pushed us to substantially improved the manuscript.
